# Engineered NIR-II fluorophores with ultralong-distance molecular packing for high-contrast deep lesion identification

Zhe Feng [1,5], Yuanyuan Li[2,5], Siyi Chen [1], Jin Li[1], Tianxiang Wu[1], Yanyun Ying [3], Junyan Zheng[3], Yuhuang Zhang[1], Jianquan Zhang[4], Xiaoxiao Fan[1], Xiaoming Yu [3], Dan Zhang [3], Ben Zhong Tang [4] ✉ & Jun Qian [1] ✉

The limited signal of long-wavelength near-infrared-II (NIR-II, 900–1880 nm) fluorophores and the strong background caused by the diffused photons make high-contrast fluorescence imaging in vivo with deep tissue disturbed still challenging. Here, we develop NIR-II fluorescent small molecules with aggregation-induced emission properties, high brightness, and maximal emission beyond 1200 nm by enhancing electron-donating ability and reducing the donor-acceptor (D-A) distance, to complement the scarce bright long-wavelength emissive organic dyes. The convincing single-crystal evidence of D-A-D molecular structure reveals the strong inhibition of the π-π stacking with ultralong molecular packing distance exceeding 8 Å. The delicately-designed nanofluorophores with bright fluorescent signals extending to 1900 nm match the background-suppressed imaging window, enabling the signal-to-background ratio of the tissue image to reach over 100 with the tissue thickness of ~4–6 mm. In addition, the intraluminal lesions with strong negatively stained can be identified with almost zero background. This method can provide new avenues for future long-wavelength NIR-II molecular design and biomedical imaging of deep and highly scattering tissues.

Fluorescence bioimaging with specific labeling is an efficient approach to catch the bio-information with high contrast, such as ultrahigh signal-to-background ratio (SBR) fluorescence imaging of exposed intraperitoneal tumor nodules[1,2]. However, the shortcomings of optical imaging will surface once the imaging signals are transferred into the deep tissues. Scattering-induced imaging background and the non-negligible autofluorescence excited by the short-wavelength light (mainly in the visible region, 360–760 nm) seriously disturb the signal extraction through deep tissues. Compared with the classical spectral range of fluorescence imaging (below 900 nm), the second near-infrared (NIR-II, 900–1880 nm) window with moderate light absorption, less photon scattering, and minimized autofluorescence in tissues gives potential opportunities for high-fidelity detection of deep signals in vivo[3–10]. The vessels, lymph, tumor, etc., have been outlined successfully by virtue of the deep-penetration of NIR-II winbdow[11–25]. Further strengthening the imaging contrast of these deep details is still full of challenges, which requires (1) improvement of detectable NIR-II signals and (2) continuous suppression of background interferences.

The demand for signal improvement has fueled the boom of diversified excellent luminophores. Wherein, the organic fluorophores

[1]State Key Laboratory of Modern Optical Instrumentations, Centre for Optical and Electromagnetic Research, College of Optical Science and Engineering, International Research Center for Advanced Photonics, Zhejiang University, Hangzhou 310058, China. [2]College of Veterinary Medicine, Jilin University, Changchun 130062, China. [3]Key Laboratory of Reproductive Genetics (Ministry of Education), Department of Reproductive Endocrinology, Women's Hospital, Zhejiang University School of Medicine, Hangzhou 310006, China. [4]Shenzhen Institute of Molecular Aggregate Science and Engineering, School of Science and Engineering, The Chinese University of Hong Kong, Shenzhen 518172, China. [5]These authors contributed equally: Zhe Feng, Yuanyuan Li. ✉e-mail: tangbenz@cuhk.edu.cn; qianjun@zju.edu.cn

usually possess good biocompatibility, efficiently facilitating the biomedical applications[8,26–32]. High-performance fluorophores are generally designed as twisted structures full of molecular rotors to suppress the undesirable intermolecular π-π interactions in the aggregated state[33–37]. On the basis of this principle, a series of highly bright NIR-II molecules with absorption and emission peaks around 700 and 1000 nm, respectively, have been successfully achieved[38–40]. However, NIR-II dyes with longer spectral responses and stronger emission intensities are still challenging to obtain and rare direct evidence of the twisted structures has been given so far.

On the other hand, to suppress the imaging background, the NIR-II imaging window has been prolonged from 900/1000 nm long-pass to 1500 nm long-pass region[41–43]. However, the restrained photon scattering is mainly recognized to account for the background decline, where the light absorption in the tissue is not used to the full. Besides the reduced photon scattering, the rising light absorption can efficiently resist the scattering disturbance in tissues; and the contrast of intravital NIR-II fluorescence imaging could be thus improved[44,45]. Recent studies have revealed that the NIR-IIx region (1400–1500 nm) with moderate light absorption is promoted as one highly-potential imaging window with minimized background[44], and yet the intensification of absorption attenuation consequently puts forward stricter requirements for luminophores.

For these reasons, aiming to deep bioimaging with higher contrast, the match between bright emitters and the background-suppressed imaging windows, serving for higher-performance NIR-II fluorescence imaging, is necessary. In this work, we propose a molecular design strategy of donor engineering toward the long-wavelength NIR-II emitters and achieve high-contrast NIR-IIx + NIR-IIb (1400–1700 nm) deep tissue imaging. To improve the brightness of fluorophores and thereby enhance the imaging signals, we enhance donor-acceptor (D-A) interactions through the reduction of D-A distance based on the guideline of "backbone distortion and molecular rotors". After rational programming, the as-synthesized 2FT-oCB molecule derived from benzo[1,2-c:4,5-c′]bis[1,2,5]thiadiazole (BBTD) shows strong emission beyond 1400 nm with an absorption peak at 829 nm (molar extinction coefficient, $\varepsilon$ of $2.3 \times 10^4 M^{-1} cm^{-1}$) and an emission peak at 1215 nm. Single-crystal structure analysis reveals the intense twist and ultralong molecular packing distance exceeding 8 Å, which can effectively avoid π-π stacking. The direct evidence of X-ray crystallography for the BBTD-cored highly-twisted dyes with long-wavelength responses is given. The solvent transfer from natural water (hydrogen oxide) to heavy water (deuterium oxide) after nanoprecipitation finally recovers the bright NIR-II emission and extends it to 1900 nm. To obtain higher contrast, we collect the tailing emission of the NIR-IIx region extra in the deep organ imaging, further suppressing the imaging background. Diverging from the traditional collection beyond 1500 nm (NIR-IIb window) where only the declined photon scattering is considered, the rising light absorption in the NIR-IIx + NIR-IIb window can attenuate the disturbance of scattering photons. The eventual SBRs of the biostructures at the depth of about ~4–6 mm reach over 100 without opening the abdomen. In addition, the intraluminal lesions containing much water, such as intrauterine residue, are simultaneously negatively stained, drawing to almost zero imaging background. This work successfully unlocks sensitive deep tissue deciphering using the engineered D-A-D fluorophores that emit strong signals in the optimal spectral window with weak imaging background.

## Results

### Donor engineering of long-wavelength D-A-D emitters

Triphenylamine-based alkylthiophene motif is a superb donor unit in highly-twisted D-A structures towards restrained intermolecular interactions, from which the general donor engineering developed in this work starts, as described in Fig. 1a. Starting with the previously reported excellent molecule 2TT-oCB[38], three dyes of 2MTT-oCB,

2MPT-oCB, and 2FT-oCB with methoxy-triphenylamine, methoxy-diphenylamine, and fluorene-based diamine, respectively, at the periphery, and an acceptor core of BBTD were designed via molecular rotor engineering, whose chemical structures and maximal absorption/emission wavelengths are visualized graphically in Fig. 1a. The strong electron-withdrawing ability of the central BBTD core can drive the absorption/emission to the long-wavelength range when coupled with electron donors. Ortho-hexyl-substituted thiophene can effectively distort the thiophene-BBTD-thiophene backbone to suppress the intermolecular π-π interactions. Some previous contributions have been made to enhance the donating ability and twisting the structures via increasing the numbers of donors[46–48] or adjusting the D-A units[49–51]. Similarly, the first step from 2TT-oCB to 2MTT-oCB increases donating ability and D-A interactions resulting from the introduction of methoxy electronic donating group. In this study, we propose a strategy of cropping the π-bridge to enhance the D-A interactions in the π–π conjugated skeleton. From 2MTT-oCB to 2MPT-oCB, the removal of the phenyl unit in triphenylamine can significantly shrink the D-A distance. Such transformation further twists the molecular structure, effectively avoiding the π-π stacking. We have to admit that the improvement of the conjugation length brings not only red-shifted spectral response but declined quantum yield (QY). Therefore, further adjustment could be conducted to maintain the emission intensity. Here, 2FT-oCB with electronic donating fluorene unit is designed to regulate the D-A interactions. On the other hand, the introduction of the planar blocks aims to improve the molar extinction coefficient. The synthetic route and structural characterization of the above molecules associated with key intermediates are displayed in Supplementary Figs. 1–19.

### Signal crystal analysis of BBTD-cored molecules

To confirm the chemical configuration, density functional theory calculations were carried out using Gaussian 09 program. The highest occupied molecular orbital (HOMO) of the molecules is delocalized along the whole molecule, revealing an excellent molecular conjugation, while the lowest unoccupied molecular orbital (LUMO) is primarily located on the BBTD core. Notably, these low energy gaps (-1.3 eV) suggest strong absorptions in the NIR biological window (see Supplementary Fig. 20). The optimized conformations in Supplementary Fig. 21 show that all the molecules adopt twisted architectures with dihedral angles of ~50°, confirming the steric hindrance between BBTD and ortho-positioned alkyl chains. Besides the theoretical calculation, the molecular conformation and packing in the crystalline state were investigated. The single crystals of 2FT-oCB were obtained by dissolving with dichloromethane and then slowly diffusing n-hexane into the solution. The crystal data and the collection parameters of X-ray crystallography are summarized in Supplementary Table 1. The monomer displayed in Fig. 1b suggests highly-twisted molecular structures essentially in agreement with the calculations. It can be seen clearly in the molecular packing state that 2FT-oCB molecules tend to be distributed in parallel with an ultralong-distance of ~8.5 Å, which efficiently reduces intermolecular interactions. The results provide strong evidence for the twisted structure design and long packing distance conception in the NIR-II D-A-D fluorophores.

### Optical response of the as-synthesized dyes

The photophysical properties of mentioned four dyes were studied by UV-vis-NIR and photoluminescence (PL) spectroscopy. The effect of enhancing D-A interactions on the dyes was investigated in the tetrahydrofuran (THF) solution. 2TT-oCB, 2MTT-oCB, 2MPT-oCB, and 2FT-oCB show generally increased maximal absorption wavelength at 695, 736, 860, and 829 nm, respectively (see Fig. 1c), with respective peak $\varepsilon$ of $1.1 \times 10^4$, $1.3 \times 10^4$, $1.8 \times 10^4$, and $2.3 \times 10^4 M^{-1} cm^{-1}$ and $\varepsilon$ of $0.5 \times 10^4$, $1.0 \times 10^4$, $1.4 \times 10^4$, and $2.2 \times 10^4 M^{-1} cm^{-1}$ at the biological window of 793 nm (see Supplementary Fig. 22). This result indicates that

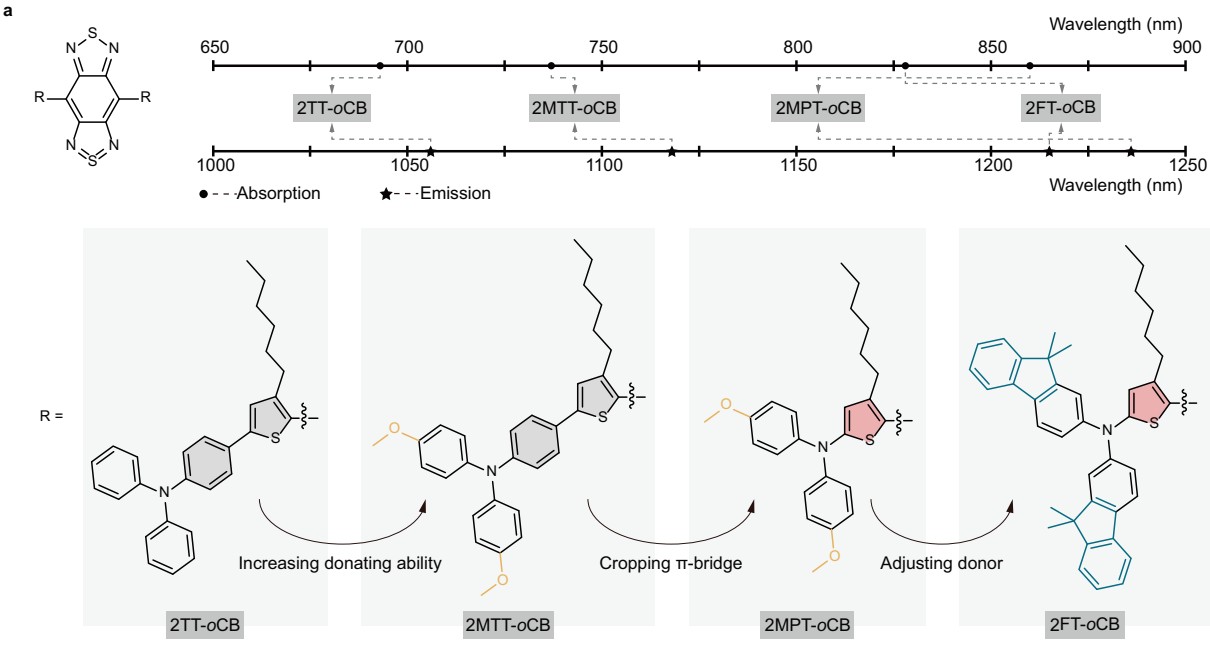

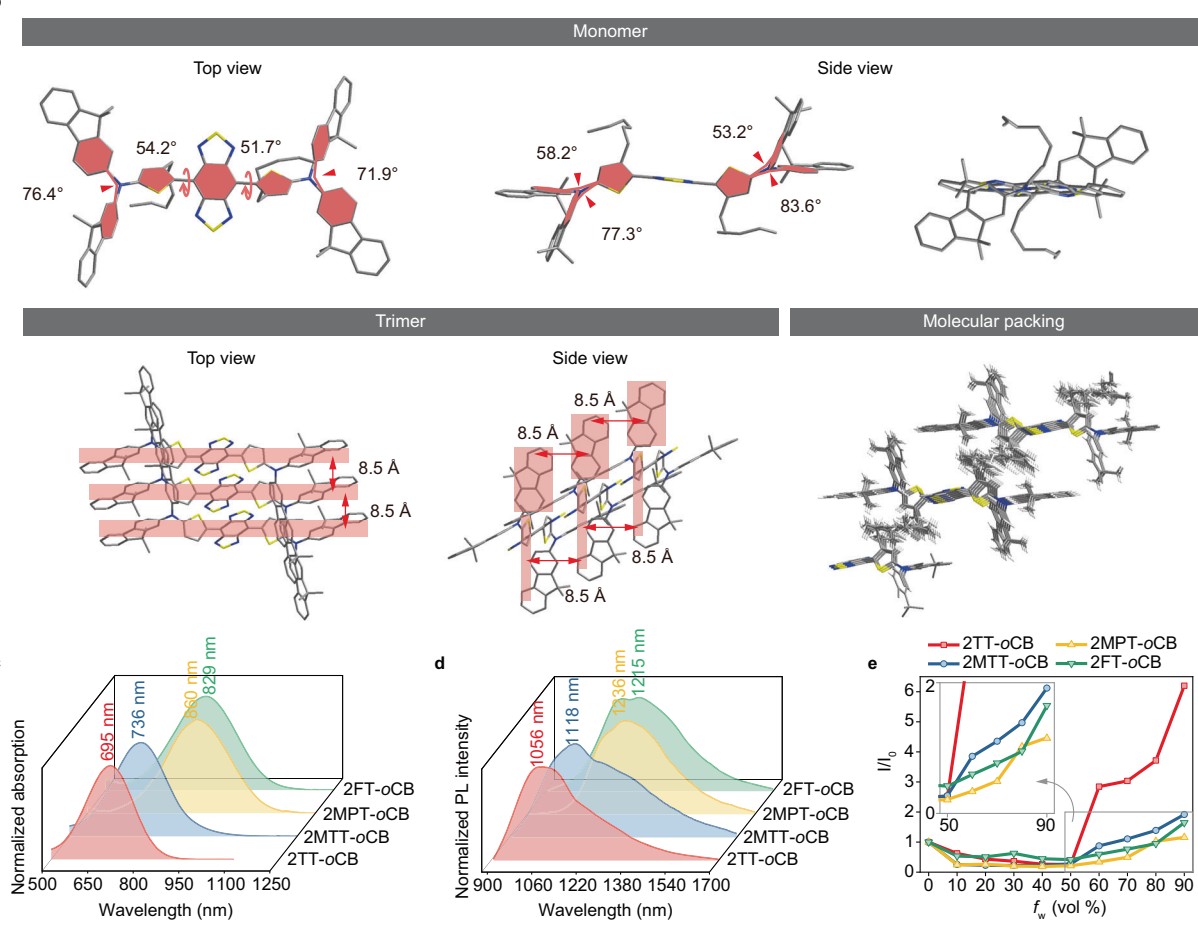

**Fig. 1 | The molecular engineering of NIR-II fluorophores with bright and wavelength-extended emission. a** The designed molecules via donor engineering and the optical responses. **b** The single-crystal structures of the 2FT-*o*CB in monomer, trimer, and molecular packing mode. **c** The normalized absorption and **d** the normalized PL spectra of the four molecules dissolved in THF. **e** The plot of the PL peak intensity of the molecules versus $f_w$. $I$ and $I_0$ represent the peak intensities in the mixture with specific $f_w$, and the pure THF ($f_w = 0$). The inserted image shows the enlarged curves with the $f_w$ from 50% to 90%.

increasing donating ability and cropping the π-bridge can not only red-shift the maximal absorption wavelength but also boost the absorption intensity. Especially, though the peak absorption wavelength assumes a slight hypsochromic shift from 2MPT-oCB to 2FT-oCB after adjusting the donor, the absorption strength is efficiently enhanced mainly owing to increasing the planarization. Notably, such a strong $\varepsilon$ ($2.2 \times 10^4 \, M^{-1} \, cm^{-1}$) of 2FT-oCB at 793 nm is favorable for excitation light to penetrate deep tissues with low photodamage. PL spectra suggest that 2TT-oCB, 2MTT-oCB, 2MPT-oCB, and 2FT-oCB display emission peaks at 1056, 1118, 1236, and 1215 nm in THF, respectively, which can be applied as NIR-II emitters for deep tissue visualization (see Fig. 1d). The red-shifted absorption and emission spectra indicate that both improving the donating ability and reducing the length of π-bridge can efficiently enhance the D-A interaction. Generally, small organic molecules always suffer fluorescence quenching dispersed from good (organic solvents) to poor solvent (water). Modifications of molecular structure for water solubility enhancement or aggregation avoidance are still challenging to keep strong emission at long wavelengths in the NIR-II region. Aggregation-induced emission (AIE) effect could be an efficient strategy in parallel to manufacture bright long-wavelength-emitting organic fluorophores[52–57]. To further investigate the fluorescence properties in the aggregated state, the PL spectra of dyes in the THF/water mixture with different water fractions ($f_w$) were carried out. As shown in Fig. 1e, the PL intensity of dyes decrease first from $f_w = 0$ to $f_w = 40\%$ due to the formation of dark twisted intramolecular charge transfer state (a dark state that can quench the fluorescence), while increase significantly with an increase of $f_w$ from 40% to 90% owing to the trigger of AIE effect via the mechanism of restriction of intramolecular motion by aggregation formation. It is noteworthy that few AIE dyes possess such long maximal emission wavelengths and the AIE molecules in this work (especially 2MPT-oCB and 2FT-oCB) can be ranked among the AIE emitters with the longest emission wavelength.

## Characterization of the nanofluorophores

To further decipher the fluorescence properties, we encapsulated the dyes into dots by the nanoprecipitation method using biocompatible surfactant Pluronic F-127 (see Fig. 2a). Except 2MPT-oCB dots (860 nm), the 2TT-oCB, 2MTT-oCB, and 2FT-oCB dots display obviously red-shifted maximal absorption wavelength as 729, 759 and 846 nm, respectively (see Fig. 2b), compared with the solution profiles (see Fig. 1c). On the other hand, the dots show fluorescence at the NIR-II region at 1032, 1092, 1204, and 1115 nm, respectively (see Fig. 2c), slightly blue-shifted than their solution state (see Fig. 1d). The excellent absorption (>840 nm) and emission (>1100 nm) properties of 2MPT-oCB and 2FT-oCB dots are particularly favorable for high-quality bioimaging. The fluorescence images of 2TT-oCB, 2MTT-oCB, 2MPT-oCB and 2FT-oCB dots with same concentrations were recorded using seven different long-pass (LP) filters (900, 1000, 1100, 1200, 1300, 1400 and 1500 nm) (see Fig. 2d). Under 900 and 1000 nm LP filters, 2TT-oCB dots display the strongest fluorescence signal owing to the high QY (8.4%) in the NIR-II region. Figure 2d demonstrates that simply increasing the donating ability from 2TT-oCB to 2MTT-oCB induces obvious decline of fluorescence intensity owing to the enhanced D-A interactions. However, by cutting the π-bridge (D-A interaction obviously increased), the wavelength of the maximal absorption/emission of 2MPT-oCB is redshifted (see Figs. 1c, 1d, 2b and 2c). Notably, 2MPT-oCB dots still maintain strong fluorescence intensity owing to the highly-twisted architecture without phenylene π-bridge (see Fig. 2d). With an increase of wavelength of filters from 1100 to 1500 nm, an overwhelming fluorescence signal is observed in 2FT-oCB dots decorated with highly-twisted skeletons and planar fluorene units, compared with the other three dots. It is noticeable that 2FT-oCB dots possess the strongest fluorescence intensity under the 1400 nm LP filter, which is ~2.5 times stronger than the previously reported

excellent 2TT-oCB dots[38]. Therefore, we chose 2FT-oCB dots from the designed four fluorophores as the example for the following in vivo imaging. With the IR-26 in the 1,2-dichloroethane (0.5%) chosen as the reference (see Fig. 2e), the QY of 2FT-oCB dots beyond 900 nm can be calculated as ~0.95%. Furthermore, the QY beyond 1400 nm can be determined as ~0.11%. As shown in Supplementary Fig. 23 and Fig. 2f, the 2FT-oCB dots coated by the F-127 possess a spherical structure, with diameters around 50 nm measured by transmission electron microscopy (TEM) and dynamic light scattering (DLS). Importantly, the 2FT-oCB dots show almost no decline of fluorescence intensity after 30 min of the continuous laser irradiation (see Fig. 2g), which demonstrates their excellent photostability.

## Natural water in the biological tissue suppresses the imaging background

The visible-NIR imaging windows are shown in Supplementary Fig. 24, which are defined according to the photon scattering and light absorption by tissue. Taking the emission spectra of 2FT-oCB dots in water and the light absorption properties of water into account (see Fig. 3a), the imaging within 1300–1700 nm of the vertical line sample (Depth = 1 mm) upon a horizontal line (Depth = 2 mm) acting as background is simulated via Monte Carlo method (see Fig. 3b). The calculated coefficients of variation ($C_v$) reveal that compared to the 1300–1700 nm and 1500–1700 nm, NIR-IIx + NIR-IIb image gives the best diffusion depression. To directly compare the in vivo performance in different imaging windows and determine the optimal long-pass detection region of NIR-II fluorescence off-peak imaging, the whole-body vessel imaging in mice was then conducted after intravenous injection of 1 × phosphate-buffered saline (PBS) solution of 2FT-oCB dots (1 mg/mL, 200 μL, normal water-based buffer solution). As shown in Supplementary Fig. 25, the strong background is suppressed with 1400LP collection, making the vessel network clearly outlined (see Fig. 3c–e). As shown in Fig. 3f, the vessel above the liver in the 1400LP fluorescence image shows the best SBR of 1.27 while the same vessel possessed the SBRs of 1.07 and 1.16 in 1300LP and 1500LP images, respectively. The spatial frequency distributions of the images in various regions can be seen in the fast Fourier transform (FFT) results (see Fig. 3g–i and Supplementary Fig. 26). Similarly, the statistical analysis exhibited a wavelength-related trend in which the intensity of high spatial frequency was positively correlated to the restrained photon scattering approximately and it emerged maxima near the light absorption peak wavelengths (Fig. 3j and Supplementary Fig. 27). Due to the intense light absorption at ~1450 nm by water, the background arising from the diffused components is highly inhibited, leading that the NIR-IIx + NIR-IIb (1400–1700 nm) image possesses more high spatial frequency even than the NIR-IIb (1500LP) image which has been long widely acknowledged with the best quality. In addition, the diameters of three selected bright vessels in Supplementary Fig. 28a–c were measured after each inverse fast Fourier transform (iFFT) via cutting the cut-off radius of the low-pass filtering in frequency domain as shown in Supplementary Fig. 28d. As shown in Supplementary Fig. 28e–g, due to the dual reduction of random noise and useful details, each vessel exhibits an optimal cut-off, where the full width at half maxima (FWHM) is minimized. Interestingly, the best FWHM of the vessels in the 1400LP image almost emerges without short-pass filtering, which fully reveals that detailed signals in the imaging with 1400LP collection account for a considerable proportion in high spatial frequency components, shrinking the disturbance of random noise.

The imaging performance of the NIR-IIx window using the water-based solution of the 2FT-oCB dots is also explored, which can be seen in Supplementary Fig. 29. Compared with the NIR-IIx + NIR-IIb imaging, the NIR-IIx imaging gives higher contrast in the visualization of superficial blood vessels. However, excellent quality of NIR-IIx imaging may sacrifice partial deep details, since the strong light absorption can

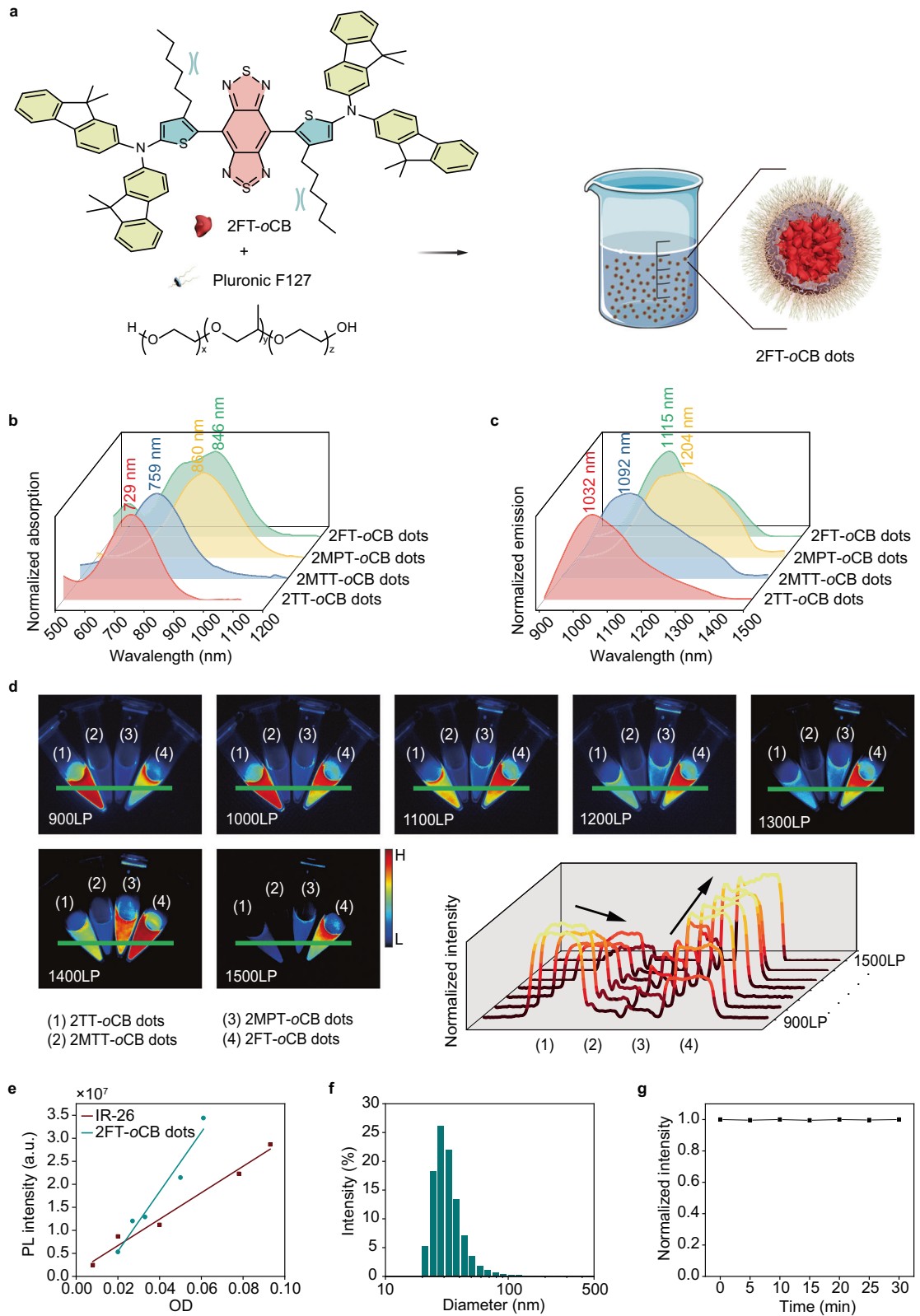

**Fig. 2 | The characterization of the compounds. a** Schematic illustration of the fabrication of fluorescent dots. **b** The normalized absorption and **c** the normalized PL spectra of the four kinds of AIE dots dispersed in water. **d** The normalized fluorescence images of tubes containing the aqueous dispersion of four fluorescent dots with the same concentration and the normalized cross-sectional fluorescence intensity profiles along the green lines. (1) 2TT-*o*CB dots; (2) 2MTT-*o*CB dots; (3) 2MPT-*o*CB dots; (4) 2FT-*o*CB dots. **e** The plots for the integrated fluorescence intensities of 2FT-*o*CB dots in heavy water and IR-26 in the 1, 2-dichloroethane at five different concentrations. a.u. here represents arbitrary units. **f** The DLS results of the 2FT-*o*CB dots. **g** The photostability test of the 2FT-*o*CB dots under continuous irradiation.

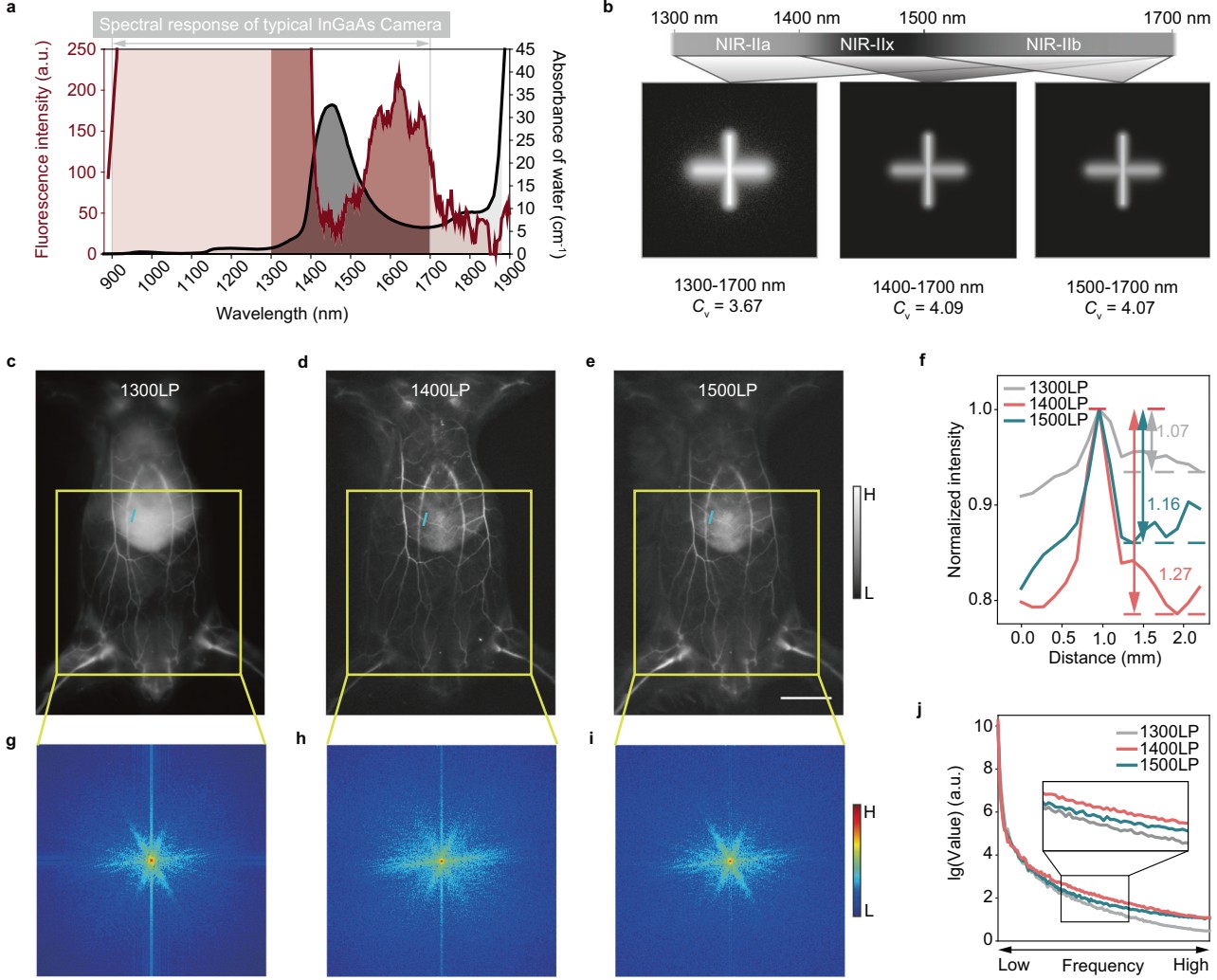

**Fig. 3 | The natural water in the biological tissues optimizes the NIR fluorescence imaging. a** The enlarged emission spectrum of 2FT-*o*CB dots and the absorbance of water in visible-NIR imaging windows. **b** The imaging simulation of the 2FT-*o*CB dots (water solution) in deep tissues via Monte Carlo method. The **c** 1300LP, **d** 1400LP and **e** 1500LP whole-body fluorescence images in the same mouse after intravenous injection of 2FT-*o*CB dots. Scale bar, 10 mm. **f** Cross-sectional fluorescence intensity profiles along the indigo lines of the blood vessel in (**c−e**). The spatial frequency maps after FFT of the **g** 1300LP, **h** 1400LP and **i** 1500LP images. The spatial frequency gradually increases outward from the center of the map, and the color bar indicates the intensity. **j** The spatial frequency distribution of (**g−i**). a.u. in **a** and **j** here represents arbitrary units.

simultaneously restrain the useful signals when suppressing the imaging background. In the imaging with rich vertical information, the NIR-IIx + NIR-IIb imaging with large depth of field is a reasonable choice, which can also better utilize the photoluminescence of the 2FT-*o*CB dots, and bring more signal as well as less background to the NIR-IIb imaging.

### Biocompatibility testing of the nanofluorophores

The whole-body imaging of mice was performed within 24 h after intravenous injection. Besides the liver, the gut is also gradually lighting up, which could be seen in Supplementary Fig. 30. Compared with the indocyanine green (ICG)-assisted vessel imaging, a longer blood circulation time could be achieved using 2FT-*o*CB dots (see Supplementary Fig. 31a, b). In the excellent 1400−1700 nm window, 2FT-*o*CB dots show better imaging quality than ICG, which could be found in Supplementary Fig. 31c.

Interestingly, after intravenous injection of 2FT-*o*CB dots, it is found that the bright NIR-II signals can be detected in the feces from the experimental groups, far stronger than those from the control groups (treated with the 1 × PBS), which is shown in Supplementary

Fig. 32. Meanwhile, the urine excreted by the mice treated with 2FT-*o*CB dots shows no difference in the fluorescence detection compared with the control groups. These results definitely demonstrate that hepatobiliary excretion is the main route rather than renal excretion. Then, we are encouraged to deeply explore the biocompatibility of the nanofluorophores including cytotoxicity and histological examinations after intravenous injection. As shown in Supplementary Fig. 33, no obvious cytotoxicity can be observed among the selected normal and tumor cell lines. To examine the safety of 2FT-*o*CB dots in vivo, two groups of Institute of Cancer Research (ICR) mice were administered with an intravenous dose of 2FT-*o*CB dots (as the experimental group) and 1 × PBS (as the control group), respectively. After nearly one month, the mice were euthanized and the histopathologic test was then performed. As shown in Supplementary Fig. 34 and Supplementary Fig. 35, slight hydropic degeneration of hepatocytes and undetectable disruption to other tissues are observed. Besides, no statistically significant differences in the organ/body weight ratios between the two groups after 28 days of injection (see Supplementary Fig. 36). Furthermore, Supplementary Fig. 37 presents the results of blood routine tests at

1 day and 28 day after intravenous injection, in which no significant fluctuation in the hematological indicators of white blood cell, red blood cell, hemoglobin and platelet is shown, compared with the control values. Importantly, the comparisons of liver function indicators of alanine transaminase and total bilirubin, and kidney function indicator of creatinine indicate that the hepatic and renal functions have not been significantly influenced (see Supplementary Fig. 38). Though the 2FT-oCB dots exhibit no obvious signs of toxicity, we still believe the safety risks can be effectively minimized without intravenous injection.

## Heavy water dispersion enhances the NIR-II signals

The absorption by O-H bonds in water results in the light absorption features with the peak wavelengths at ~980 nm, ~1200 nm, ~1450 nm, and ~1930 nm. In the whole NIR-II region, the absorption at ~1450 nm is most pronounced. Thus, the growing light attenuation from 1400 nm would inevitably deplete the emission of the fluorophores dispersed in water. The typical absorption spectra of water and heavy water given in Supplementary Fig. 39a reveal that the fluorescence loss in the NIR-II region induced by solvent could not be ignored anymore. Accordingly, the dispersion of 2FT-oCB dots in heavy water is able to recover the emission loss near 1450 nm. After ultrafiltration, the heavy water dispersion was obtained by re-diluting the concentrated 2FT-oCB dots. The PL spectrum of 2FT-oCB dots expands in the heavy water dispersion and the peak emission wavelength is red-shifted by 30 nm (from 1115 nm to 1145 nm), which is shown in Supplementary Fig. 39b. Besides, the enlarged spectrum exhibits a highly efficient fluorescence recovery beyond 1400 nm. After calculation, the proportion of the NIR-IIx emission to the NIR-IIx + NIR-IIb emission rises sharply, up to 63.0% from 16.1% after heavy water redispersion. The emission intensity in the whole NIR-II region of the dots dispersed in heavy water is twice that of dots dispersed in water. Moreover, the emission in heavy water beyond 1400 nm was calculated to be ~20 times that in water. We further calculated the attenuated emission results using the emission spectrum of 2FT-oCB dots in the heavy water and light absorption properties of water. After tuning the parameters, a fitted attenuation process was established with 95% confidence bounds, and a credible simulation data with photon restraining in water was obtained and shown in Supplementary Fig. 40 with the name of calculated data. Compared with the measured spectrum of 2FT-oCB dots in water, it can be seen that the severe emission loss beyond 1400 nm of the calculated data and the measured data is consistent and the blue-shifted peak emission wavelength difference is just 3 nm (1118 nm of the calculated data and 1115 nm of the measured intensity in water, which may be caused by the simulation errors), indicating the main role played by water in the emission differences of 2FT-oCB dots in the two solvents.

Considering the emission recovery of the 2FT-oCB dots from water to heavy water, we again simulated the imaging through the deep tissues in three windows using the emission properties of the heavy water solution of 2FT-oCB dots. The results shown in Supplementary Fig. 41 confirmed the background suppression strength of the NIR-IIx + NIR-IIb window. The Intralipid® phantom study was then conducted to mimic the hindrance of light propagation in biological tissues. As shown in Supplementary Fig. 42, the imaging with 1400LP collection shows the best FWHM. The seemingly abnormal results can be attributed to the positive contribution of light absorption[44,45]. In addition, compared with the water dispersion, Supplementary Fig. 43 shows that heavy water dispersion can improve the SBR because of the signal enhancement. The fluorescence recovery via re-dispersing 2FT-oCB dots in heavy water is believed to be an effective and essential approach to the deep-buried hollow organ deciphering, with organ perfusion instead of direct injection into the internal environment.

## High-contrast in vivo fluorescence colonography

Computed tomography colonography has been widely utilized in clinics to examine the large intestine for cancer and growths called polyps by special X-ray equipment. The resulting ionizing radiation becomes the necessary cost to obtain an interior view of the colon which is otherwise only seen with a more invasive procedure by virtue of an endoscope. The fluorescence colonography was conducted in different imaging windows of the NIR-II region after colonic perfusion of 2FT-oCB dots (1 mg/mL, 200 μL, deuterium oxide dispersion). As shown in Supplementary Fig. 44 and Fig. 4a, integrated interaction between light and tissue in different imaging regions presents characteristic performance. Different degrees of diffused components blend in with the ballistic light, and they are both collected onto the near-infrared sensitive camera eventually. Some blurry edges and expanded outlines will undoubtedly interfere with the judgment and the operation. Not so long ago, one imaging case about the tissue labeling with ultrahigh SBR value of ~48.5 was achieved, in which the mouse abdomen was opened and thus there was no thick tissue covered[1]. Encouragingly, as shown in Fig. 4b, the proposed non-invasive imaging mode with NIR-IIx + NIR-IIb detection gives a comparable SBR (~46.6) close to 50 even at the depth of 3–6 mm with serious tissue disturbance (see Supplementary Fig. 45). To quantitatively divide the targeted area from the imaging background, the selected original results were processed by binarization, using three threshold values (TH = 0.27, 0.32 and 0.37, representing that 27%, 32%, and 37% of the highest brightness in the selected area were set as the threshold values for the signals/background determination), as shown in Fig. 4c. The overall trend is that the calculated area shrank with the red-shifting of the imaging wavelength (see Fig. 4d). Notably, the selected section of the colon is minimized in 1400–1700 nm. Meanwhile, the area SBR can be determined with the mean intensities of the segmented colon calculated as the signals, and the mean intensities of the rest regarded as the background. It can be patently seen in Fig. 4d that the SBR value reaches the maximum with 1400LP collection. The area SBR analysis may offer a precise evaluation of the deep-buried organs, since it considers all the signal and background pixels. Eventually, the 2FT-oCB dots can be excreted from the mouse body, and the feces in the bright field image and the NIR-II fluorescence image under 793 nm laser excitation can be seen in Supplementary Fig. 46a, b. The bright field and NIR-II fluorescence images after opening the abdomen are shown in Supplementary Fig. 46c and Supplementary Fig. 47, and ensure the labeling of the colon. Meanwhile, the results of the histology study of the colons show no obvious side effects after the perfusion of 2FT-oCB dots in heavy water dispersion (see Supplementary Fig. 48).

## High-consistency in vivo fluorescence cystography

Bladder disorders are common as we age, which influence our health in the urinary system. The observation of bladder morphology can assist diagnose the bladder rupture, tumor, and diverticulum to some degree. NIR-II fluorescence cystography through the skin and muscle was conducted after intravesical instillation of the 2FT-oCB dots (1 mg/mL, 20 μL, deuterium oxide dispersion). The non-invasive NIR-II fluorescence image of bladder was set into a green channel. (see Fig. 4e). As shown in Fig. 4f, the line SBR of the NIR-IIx + NIR-IIb fluorescence bladder image reaches ~46.5 with the tissue depth of 2–3 mm (see Supplementary Fig. 49). The binary and segmented bladder images in Fig. 4g and Supplementary Fig. 50 indicate that the NIR-IIx + NIR-IIb spectral window minimizes the scattering disturbance. In addition, the bladder's area and area SBR calculations in Fig. 4h also demonstrate the 1400–1700 nm gives the best imaging performance. In addition, the bladder image after opening the abdomen was set into a red channel for comparison, and then the green and red images were merged into one and the areas with high-consistency were presented in yellow. As shown in Fig. 4i, the near-infrared

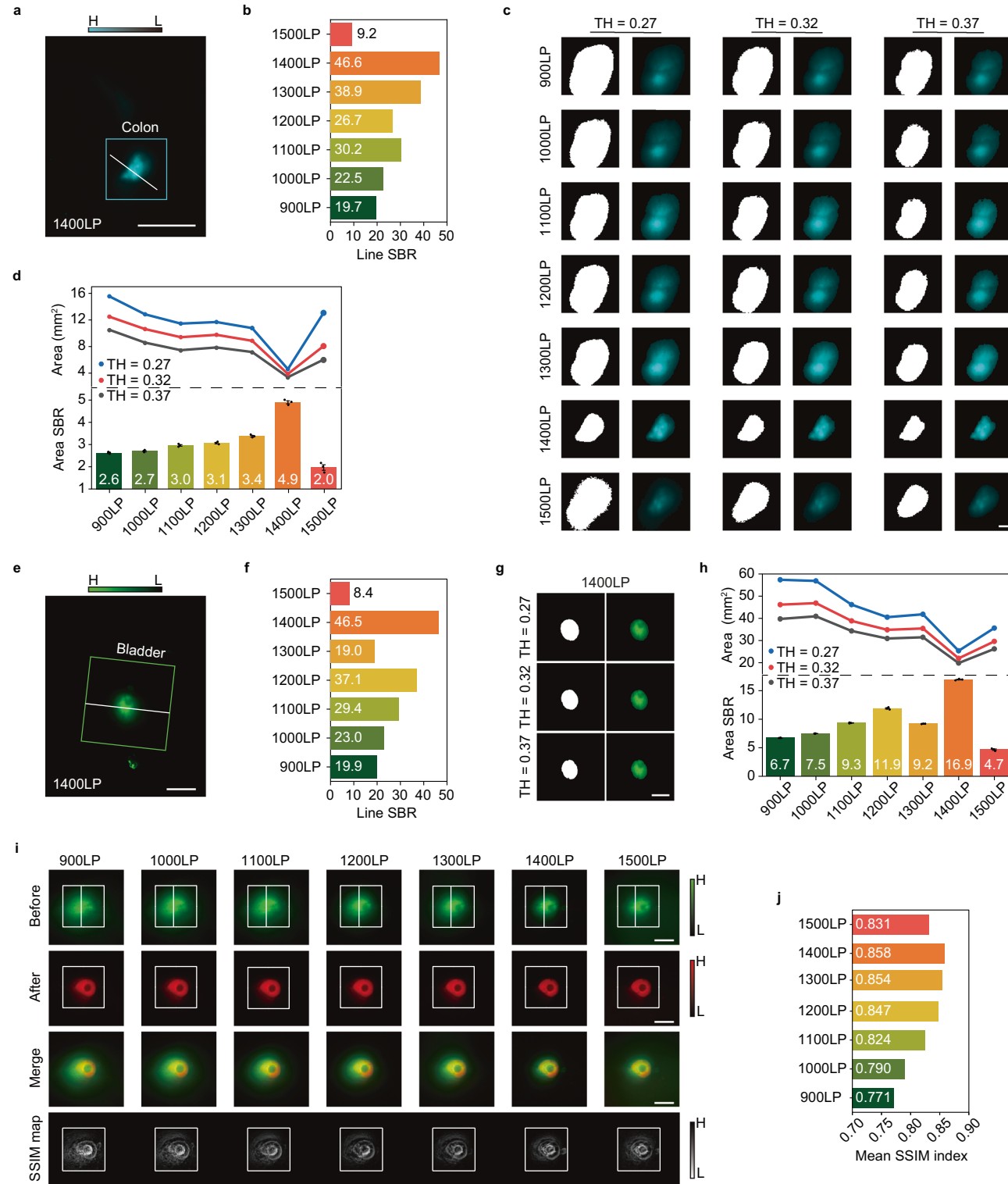

**Fig. 4 | High-contrast in vivo colon and bladder imaging. a** A typical NIR-IIx + NIR-IIb fluorescence colonography image. Scale bar, 5 mm. **b** Line SBR results along the white lines in (a) and Supplementary Fig. 44. **c** The binary and segmented fluorescence colon images of white squares in Supplementary Fig. 44 with the TH of 0.27, 0.32 and 0.37. Scale bar, 1 mm. **d** The signal areas after binarization and the area SBRs of the colon images in different spectral regions. The statistical analysis is conducted using three segmented images with three TH values ($n = 3$) and the mean area SBRs are shown in the columns. Error bars indicate standard error of the mean (SEM). **e** A typical NIR-IIx + NIR-IIb fluorescence cystography image. Scale bar, 5 mm. **f** Line SBR results along the white lines in (i). **g** The binary and segmented NIR-IIx + NIR-IIb fluorescence bladder images with the TH of 0.27, 0.32 and 0.37. Scale bar, 5 mm. **h** The signal areas after binarization and the area SBRs of the bladder images in different spectral regions. The statistical analysis is conducted using three segmented images with three TH values ($n = 3$) and the mean area SBRs are shown in the columns. Error bars indicate SEM. **i** The images of fluorescence cystography before and after opening the abdomen, the merged images processed via image J and the SSIM maps output from MATLAB. Scale bars, 5 mm. **j** The SSIM index of the seven SSIM maps in different collection spectral regions.

detection here was progressively red-shifted to restore the original appearance of biostructure as much as possible. The accurate quantification by structural similarity index measure (SSIM) can assist the image quality assessment, Therefore, the SSIM maps are given in Fig. 4i in the corresponding imaging window as complementary evaluation for the merged image with quantification. The merged bladder image in 1400–1700 nm shows the best matching, and the presentation exceeds the performance in the NIR-IIb region. The mean SSIM indexes in the whole SSIM maps are shown in Fig. 4j. With the prolonging of the imaging window, the SSIM index gradually improves and peaks at the map in 1400–1700 nm. The highest consistency with the NIR-IIx + NIR-IIb detection in the NIR-II non-invasive visualization can be concluded. As expected, the urine subsequently excreted from the mouse exhibits bright NIR-II emission, as shown in Supplementary Fig. 51a, b. After opening the abdomen, it can be seen that the bladder of the mouse was bulging with 2FT-*o*CB dots of mauve (see Supplementary Fig. 51c). The hematoxylin and eosin (H&E) staining results of the bladders in Supplementary Fig. 52 show that there exists no obvious difference between the groups perfused with the heavy water dispersion of 2FT-*o*CB dots and the 1 × PBS, respectively. Besides, the good stability of 2FT-*o*CB dots in the urine was confirmed, which could be found in Supplementary Fig. 53. Supplementary Fig. 54 gives a typical case of intraoperative bladder injury, where the perfused 2FT-*o*CB dots leak into the abdominal cavity and light up the whole body. The in vivo NIR-IIx + NIR-IIb fluorescence imaging technique with high contrast and consistency is believed to possess tremendous translational potential in future clinical scenes.

## Deep-penetration in vivo fluorescence hysterography

Fluorescence hysterography is free of ionizing radiation but is admittedly challenging since the uterus is always deep in the body and there usually exist adipose tissues surrounding uteruses and blocking the propagation of the optical signals, which raises the requirement of enough emission intensity for the nanoprobes. After uterine perfusion of the 2FT-*o*CB dots (1 mg/mL, 200 µL, deuterium oxide dispersion), the anesthetized mice laid on the imaging stage with the lower abdomen exposed to the irradiation of 793 nm continuous wave (CW) laser. The NIR-II fluorescence uterus images are displayed in Supplementary Fig. 55. Figure 5a gives the NIR-IIx + NIR-IIb image and the line SBR analysis in Fig. 5b confirmed that the 1400-nm LP image gives the best imaging quality even with the tissue thickness of 4–6 mm (see Supplementary Fig. 56). Also, the 1400–1700 nm window provides the most precise area segmentation and the highest area SBR value, as shown in Supplementary Fig. 57 and Fig. 5c. In addition, the calculated diameter of the right uterus is minimized at 1.69 mm in the NIR-IIx + NIR-IIb image by the measurement of FWHM, which is demonstrated in Fig. 5d and Supplementary Fig. 55, while the FWHM measured in NIR-IIb (1500LP) image was 1.78 mm. The visible brightness difference between the left and right can be blamed on the uneven distribution of the adipose and muscular tissues. After the enhancement on image brightness, it can be seen that the left uterus is also visualized and precisely positioned (see the inset images of Supplementary Fig. 55a). As shown in Supplementary Fig. 58a, b, the 2FT-*o*CB dots can pass out of the body with the constant peristalsis of the uteruses after imaging, and residual 2FT-*o*CB dots can also be removed by uterine lavage. Further demonstration about the labeling of uteruses can be seen in Supplementary Fig. 58c and Supplementary Fig. 59, and the bright 2FT-*o*CB dots from the uteruses exhibit strong NIR-II emission after opening the abdomen. Thus, the 2FT-*o*CB dots assisted in vivo fluorescence imaging with NIR-IIx + NIR-IIb detection is verified to provide precise spatial resolution and high-contrast with no invasion. Besides, the uterus treated with the heavy water dispersion of 2FT-*o*CB dots shows no noticeable damage in the H&E stained tissues (see Supplementary Fig. 60).

## Precise intrauterine residue detection with almost zero background

After delivery, abortion, or surgery of suction and curettage, there might be some intrauterine residue remaining inside the uteruses, causing hemorrhage or infection. B-scan ultrasonography is widely used in the clinic to diagnose whether there is any residue, however, the small fragments remaining in the uterus are still difficult to identify. In parallel, NIR-II fluorescence imaging can give some new ideas for accurate detection with optical resolution. As mentioned above, within the spectral region around the absorption peak, the light absorption would deplete the scattered photons. Moreover, the residual tissues are always abundant in water, which leads to strong negative staining around the bright signals due to the light absorption. Thus, the dual contribution could provide a sharp contrast for further diagnosis. As shown in Fig. 5e, the super-absorbent resin is filled into the uterus cavities, acting as the tiny foreign bodies, such as hydatidiform moles. The in vivo imaging was conducted after abdominal suture and intrauterine injection of the 2FT-*o*CB dots (1 mg/mL, 200 µL, deuterium oxide dispersion) (see Fig. 5f), and the results can be seen in Fig. 5g. With the uterine peristalsis and auxiliary pressing, the tissue depth (about several millimeters, mainly consisting of the muscle and fat tissues), as well as the tissue interruption, reduces, and the intrauterine foreign bodies are increasingly identified. To better mimic the endogenous residue, a batch of pregnant mice were obtained (see Fig. 5h). After intrauterine injection of the 2FT-*o*CB dots (1 mg/mL, 500 µL, deuterium oxide dispersion), the fetuses in the uteruses with strong negative staining can be clearly presented (see Fig. 5i–k). The 1300LP, 1400LP and 1500LP images were put into three channels with the pseudo-color enhancement of red, green, and blue, respectively. As shown in Fig. 5l, the merged image of the three channels shows generally white integrated by the three primary colors. However, the farther away from the center of the overlapped structure, the merged color gradually deepens from white (or mauve) to purple, resulting from the excess of red and blue channels on the outer contours of some pregnancy tissues (especially in the yellow frame). The interesting results here can be blamed on the relatively high light diffusion in the 1300LP and 1500LP channels. As shown in Fig. 5m, the placentas can be accurately outlined with NIR-IIx + NIR-IIb detection. Then, the detection of missed abortion was further performed. On 6th day post injection of lipopolysaccharide (LPS, a kind of endotoxin, utilized to stimulate uterus to contract, thus leading to abortion), the mice were imaged after intrauterine perfusion of the 2FT-*o*CB dots (1 mg/mL, 200 µL, deuterium oxide dispersion). From the NIR-IIx + NIR-IIb images in vivo shown in Fig. 5n, a fetus can be observed in the right uterus and some residual pregnancy tissues can be precisely detected in the left uterus. Since the residual pregnancy tissues contain much water, the tissue background is efficiently suppressed in the NIR-IIx + NIR-IIb window, leading to strong negative staining. According to the analysis in Fig. 5n, the calculated SBR comes to over 100. Some cases have reported the bioimaging possessing SBR over 100 with the background selected outside the tissue[18]. However, to the best of our knowledge, there are few precedents about such high SBRs when the signal and background are both determined in the targeted tissue. After slight auxiliary pressing and appropriate position adjusting, the fetus in the right uterus becomes clearer from Fig. 5n to Fig. 5o, due to the amelioration of the tissue obstruction. As shown in Fig. 5o, the valley value represents the lowest intensity of the negatively stained fetus, while the peak value shows the bright NIR-IIx + NIR-IIb signal labeling of the uterine cavity. When the peak signal of the fetus reaches 255 (the largest value in the 8-bit image), the valley value is measured as 0, which means the SBR value ascends to positive infinity. By positioning the boundary, the residual pregnancy tissue and the fetus are determined with a diameter of 2.2 mm (see Fig. 5n) and 5.7 mm (see Fig. 5o), respectively. Comparing the

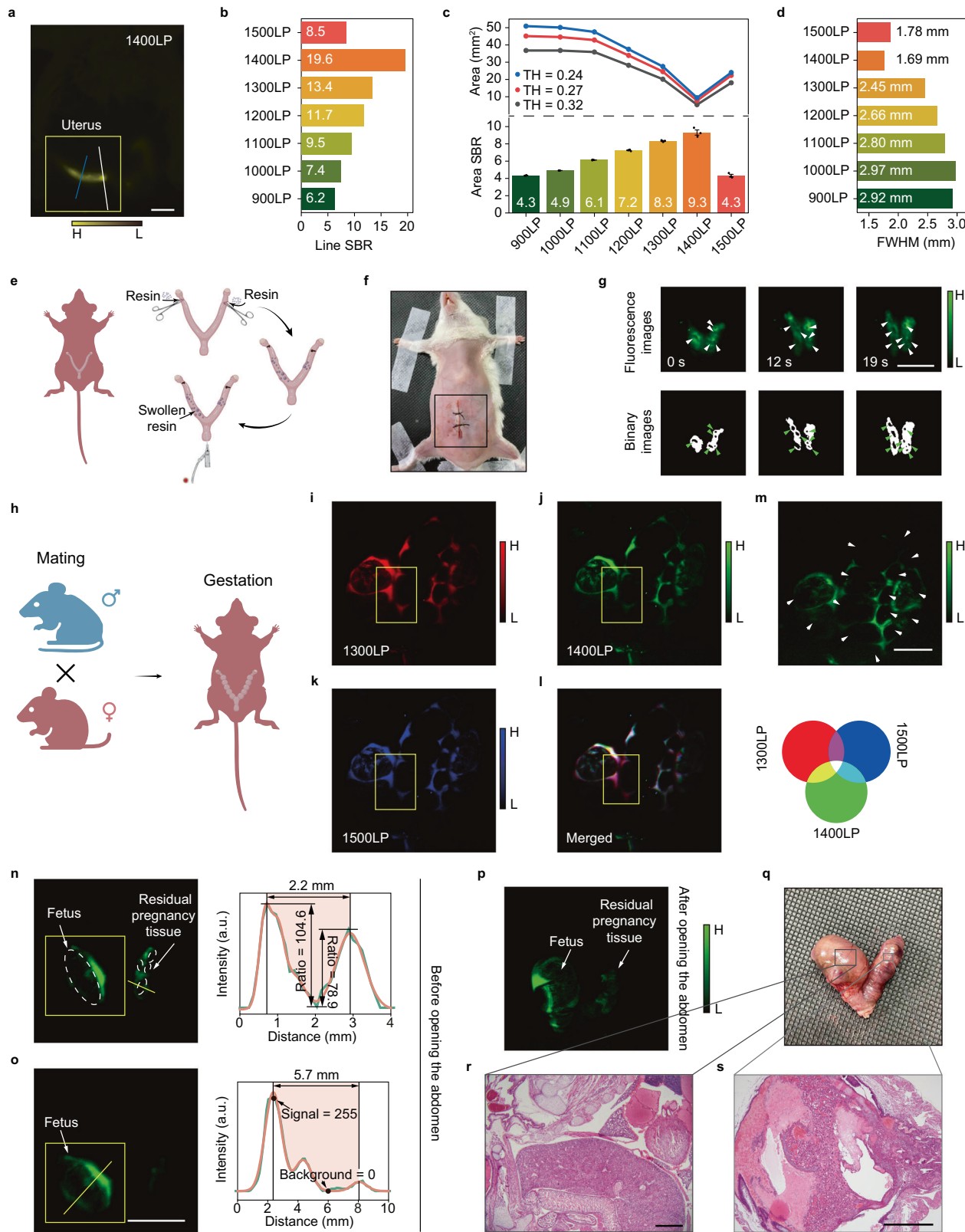

image before (see Fig. 5n, o) and after (see Fig. 5p) opening the abdomen, the hindrance of skin and adipose tissue does not disturb the precise diagnosis with NIR-IIx + NIR-IIb detection. It is worth noting that the power density (~120 mW/cm²) of the 793 nm CW laser irradiation is below the safety threshold, according to the maximum permissible exposure recommended by the American National Standards Institute (#13997). Finally, the uteruses were further

dissected out for section and H&E staining (see Fig. 5q). The section image of the right uterus shows a complete structure of the mouse fetus (see Fig. 5r and Supplementary Fig. 61), while the left side contains a mess of residual pregnancy tissue (see Fig. 5s and Supplementary Fig. 61), which is consistent with the in vivo NIR-IIx + NIR-IIb observation. The efficient labeling of fluorophores with bright NIR-IIx + NIR-IIb emission and the strong negative staining around

**Fig. 5 | The precise detection of intrauterine foreign body in vivo. a** A typical NIR-IIx + NIR-IIb fluorescence hysterography image. Scale bar, 5 mm. **b** Line SBR results along the white lines in (a) and Supplementary Fig. 55a. **c** The signal areas after binarization and the area SBRs of the uterus images in different spectral regions. The statistical analysis is conducted using three segmented images with three TH values ($n = 3$) and the mean area SBRs are shown in the columns. Error bars indicate SEM. **d** The FWHMs in different collection spectral regions along the blue lines in (a) and Supplementary Fig. 55a. **e** The operation illustration for the intrauterine foreign body detection. **f** The picture of the mouse after operation with the abdomen sutured. **g** The NIR-IIx + NIR-IIb fluorescence images with the peristalsis and manual pressing and the binary images with the black spots representing the foreign bodies. The white arrows and the green arrows point out the foreign bodies. Scale bar, 10 mm. **h** The illustration of pregnancy. The fluorescence uterine images of pregnant mouse with **i** 1300-nm, **j** 1400-nm and **k** 1500-nm LP detection. **l** The merged image of red (1300LP), green (1400LP) and blue (1500LP) channels. **m** The 1400LP image where the fetuses are marked with white arrows. Scale bar, 10 mm. **n** The fluorescence images of the missed abortion model before opening the abdomen and the cross-sectional fluorescence intensity profile along the yellow line of the residual pregnancy tissue. a.u. here represents arbitrary units. **o** The fluorescence images and the cross-sectional fluorescence intensity profile along the yellow line of the fetus. Scale bar, 10 mm. The Gaussian fits to the profiles are shown by the brown lines in (**n**) and (**o**). a.u. here represents arbitrary units. **p** The fluorescence image of the missed abortion model after opening the abdomen. **q** The pictures of the isolated uteruses. Representative histological images of the **r** right uterus (containing fetus) and **s** left uterus (containing residual pregnancy tissue) in three independent experiments. Scale bar in (**r**), 1 mm; Scale bar in (**s**), 500 μm.

the light absorption peak compose the dual visualization, which is believed to provide a powerful platform for the precise diagnosis of intrauterine residue.

## Discussion

Fluorescence imaging had long developed a reputation as a technological laggard in deep penetration. The emergence of NIR-II fluorescence imaging brought an infusion of hope to outline the deep details with high-contrast in vivo to a certain extent, but the long-wavelength bright NIR-II fluorophores towards a suitable imaging window are still rare.

To ensure both long absorption/emission wavelength and high fluorescence intensity in the aggregate state, a molecular design strategy of enhancing D-A interactions including reducing D-A distance and increasing electron-donating ability based on the design of "backbone distortion and molecular rotors" was proposed. The designed fluorophores show strong absorption properties and bright emission extending to 1900 nm. It needs to be mentioned that the 1700-1880 nm with moderate light absorption and restrained photon scattering has been recognized as the NIR-IIc subwindow[44]. The long-wavelength emission tailing of 2FT-oCB dots in the NIR-IIc region is benefit for deep-penetration especially through the adipose tissues. Importantly, we complete the single-crystal analysis of the BBTD-cored NIR-II dyes, and the distribution with an ultralong-distance of ~8.5 Å achieves one of the longest molecular packing distances. In the future, incorporating the planar blocks and adjusting the acceptors (such as replacing the BBTD core and inducing the Se atoms) can be utilized as potential programs for enhancement of long-wavelength fluorescence emission.

To efficiently restrain the imaging background, the optimum fluorescence imaging window with 1400LP (NIR-IIx + NIR-IIb) detection is verified via multiple evaluation methods. Except for the bladders, the structures and profiles of colons and uteruses would inevitably change after opening the abdomen, which makes it difficult to obtain the original morphology characteristics. So, the SSIM analysis has not been utilized in all image evaluation processes. The lesions containing much water can be further negatively stained, thus leading to the precise diagnosis with ultrahigh SBR over 100 or even zero background.

Supplementary Table 3 gives a direct comparison of our previously developed NIR-II dyes towards specific application windows, clearly indicating the superiority of the 2FT-oCB dots. To further show the advantages of the developed fluorophores in this work aiming at long-wavelength fluorescence imaging (such as NIR-IIb imaging, beyond 1500 nm), the detailed information among the excellent NIR-IIb fluorophores developed by our group and other scientists has been listed in Supplementary Table 4. 2FT-oCB dots can be listed as one of the few fluorophores that simultaneously achieve long-wavelength optical responses, strong light absorption, and high QY beyond 1400/1500 nm.

Besides the AIE fluorophores mentioned in this work, some other excellent luminophores, including but not limited to the semiconducting polymers, near-infrared quantum dots, and lanthanide-doped downshifting luminescent nanocrystals, could also be potential candidates for long-wavelength biomedical fluorescence imaging near the absorption wavelength at around 1450 nm. This method is believed to store a high potential for clinical translation considering the intraluminal perfusion instead of direct injection into the circulatory system excludes the major safety concerns of exogenous agents. Besides, the stable deuteration of water could efficiently red-shift the light absorption with no radiological concerns introduced[58]. We believe our findings presented in this work will be transformational for further fluorophore preparation and biomedical research.

## Methods

### Hydration and PEG-encapsulation of fluorescent molecules

A solution containing 3 mg of 2TT-oCB/2MTT-oCB/2MPT-oCB/2FT-oCB, 15 mg of Pluronic F-127, and 1.5 mL of THF was added into 10 mL of deionized water. The mixture was then evenly stirred in the fume hood overnight to remove the THF thoroughly. The water dispersion of 2TT-oCB/2MTT-oCB/2MPT-oCB/2FT-oCB dots was then passed through a filter with the aperture of 0.22 μm for degerming and preliminary filtration. Then, after ultrafiltration, the dots were further purified and concentrated. The heavy water dispersion of dots was obtained by redispersion in heavy water after ultrafiltration.

### Animal handling

All experimental procedures were approved by Animal Use and Care Committee at Zhejiang University. The BALB/c nude mice (~20 g) and ICR mice (~20 g) involved in this work were provided from the SLAC Laboratory Animal Co. Ltd. (Shanghai, China). All the experimental animals were housed under standard conditions in the Laboratory Animal Center of Zhejiang University, which were maintained and bred in specific pathogen-free conditions with a 12 h light and 12 h dark cycle, 25 °C room temperature and 50.0 ± 5.0% humidity and had access to food and water ad libitum. Female mice were used in the studies of fluorescence hysterography and intrauterine foreign body detection and there was no other sex bias in the animals used.

### NIR-II fluorescence macro imaging in vivo

All the mice were anesthetized before the imaging via tribromoethanol (10 mL/kgBW of the 1.25% solution). After vesical, uterine, or colonic perfusion, the deep bladder/uterus/colon fluorescence imaging was conducted with the corresponding organ exposed under the excitation of 793 nm CW laser (Suzhou Rugkuta Optoelectronics Co., Ltd., China). The dosages of 2FT-oCB dots (1 mg/mL) for the cystography, hysterography, and colonography were 20 μL, 200 μL, and 200 μL, respectively. The 900 nm, 1000 nm, 1100 nm, 1200 nm, 1300 nm, 1400 nm, and 1500 nm long-pass optical filters were purchased from Thorlabs. The simple schematic diagram of imaging system is shown in Supplementary Fig. 62. The NIR-II signals from the 2FT-oCB dots in the labeled organs are collected by the NIR antireflection fixed focus lens

with a focal length of 35 mm (TKL35, Tekwin) through the optical filters and detected by the electronic-cooling 2D (640 pixels × 512 pixels) InGaAs camera (SD640, Tekwin). The power densities of the excitation and the exposure times in the in vivo experiments were listed in Supplementary Table 2. The fluorescence images were collected from the customized software of TEKWIN SYSTEM (SD640). Quantitative analysis of the fluorescence images was performed using Image J software (Version 1.6.0, National Institutes of Health, USA). Data were analyzed using MATLAB R2020b, OriginPro 2018 (64-bit) and GraphPad Prism 8.0 (USA).

### Image processing with the fast Fourier transform and inverse fast Fourier transformation

Two-dimensional fast Fourier transformation was utilized to acquire spatial frequency maps of selected areas in whole-body vessels imaging by MATLAB R2020b. The low-frequency components were placed in the four corner parts of the FFT image while the high-frequency components were shown near the center at first. To facilitate the observation, the low-frequency components were then shifted to the center and the high-frequency components were set to the four corners. The final image was processed by the natural logarithm. The whole image processing was shown in Supplementary Fig. 63.

The zero-frequency point in spatial frequency maps was set as the center of the circle and then statistical analysis was conducted by counting the values of all pixels in the specific radius, which represented the intensity of the specific spatial frequency in the original fluorescence images. A circular low-pass filter was designed to filter out the high-frequency components. The cut-off radius was adjusted to control cut-off frequency. Then inverse fast Fourier transformation was adopted to obtain the denoising images of the whole-body vessels.

### Image processing of the SSIM assessment

Some important perception-based facts including structural degradation, luminance masking as well as contrast masking collaborated in the SSIM. Taking the luminance ($l$), contrast ($c$), and the structure ($s$) into consideration, the expression for SSIM index could be described as:

$$l(x,y) = \frac{2\mu_x\mu_y + C_1}{\mu_x^2 + \mu_y^2 + C_1} \tag{1}$$

$$c(x,y) = \frac{2\sigma_x\sigma_y + C_2}{\sigma_x^2 + \sigma_y^2 + C_2} \tag{2}$$

$$s(x,y) = \frac{\sigma_{xy} + C_3}{\sigma_x\sigma_y + C_3} \tag{3}$$

$$\text{SSIM index} = [l(x,y)]^\alpha \cdot [c(x,y)]^\beta \cdot [s(x,y)]^\gamma \tag{4}$$

In the SSIM index, $l(x,y)$ represents the comparison of the two images on brightness, $c(x, y)$ differs the two images on contrast, and $s(x, y)$ distinguishes the two images on the structural similarity and dissimilarity, where the $\mu_x$ and $\mu_x$ are the local means of intensity, $\sigma_x$ and $\sigma_y$ are the standard deviations and $\sigma_{xy}$ is the cross-covariance for images $x$ and $y$ sequentially in the images. The constants $C_1$ and $C_2$ can be given directly from MATLAB to avoid the zero denominators, and $C_2 = 2C_3$. The presented NIR-II fluorescence bladder images were caught in their respective optimal shooting scene. The weight coefficient of brightness $\alpha$ was set to zero to exclude the systematic error caused by the experimental condition since the NIR-II fluorescence image brightness taken here contained the integrated influence of different exposure time and the discrepant emission intensity in the corresponding wavelength band under specific excitation. In

comparison, the contrast and structure were not much impacted by the recording difference, so the weight coefficients of contrast ($\beta$) and structure ($\gamma$) were set to 1 empirically. Thus, the SSIM index in this work could be reduced as follows:

$$\text{SSIM index} = c(x,y) \cdot s(x,y) = \frac{2\sigma_{xy} + C_2}{\sigma_x^2 + \sigma_y^2 + C_2} \tag{5}$$

By invoking the built-in function in MATLAB R2020b, the SSIM map of the bladder image was generated, and the mean SSIM index of the whole map was calculated. SSIM map showed SSIM indexes of every pixel in the fluorescence image. In the SSIM assessment, the fluorescence bladder images through the skin, muscle, and adipose layer were set as the measured objects, and the bladder images after opening the abdomen were regarded as the references.

### Image binarization, segmentation and area SBR calculation

The selected areas in the fluorescence colon/bladder/uterus images were binarized with a given threshold value (TH) by invoking the built-in function in MATLAB R2020b. The TH, representing the proportion of the highest brightness in the selected area, determined the boundary of the signals and background, which excluded the noise disturbance to a certain extent. The areas of signals were calculated by counting the number of pixels in the binarized signal regions. The mean intensity of the binarized signals region was divided by the mean intensity of the binarized background region to calculate the area SBR.

### FWHM analysis and line SBR calculation

The cross-sectional fluorescence intensity profiles along the lines over the imaging object were acquired by Image J. Nonlinear curve fitting of the intensity profiles was executed by the Gauss fitting in OriginPro 2018C, giving the calculated FWHM. In the line SBR calculation, the peak of the fitted Gauss curve was recognized as the signal value, while the baseline of the fitted curve was considered as the background value.

### Preparation of the animal model for intrauterine foreign body detection

Female mice were anesthetized and the abdominal hair was removed. First, each mouse was placed in the supine position for laparotomy. Next, the right uterus was isolated and an incision was made in the right uterine horn. The uterine cavity was then filled with super-absorbent resin through the incision. After that, the uterus and abdomen were sutured and 200 μL PBS was injected into the uterine cavity through the cervix to swell the resin. About 30 min later, the deuterium oxide dispersion of 2FT-oCB dots (1 mg/mL, 200 μL) was perfused into the uterus for fluorescence hysterography.

Another group of females were injected intraperitoneally with 10 IU/mouse of pregnant mare's serum gonadotropin (PMSG) and received 10 IU/mouse intraperitoneal injection of human chorionic gonadotropin (HCG) 48 h later. Then they were caged with males (two females per male). The vaginal plug was checked the following morning to indicate the day 0.5 of gestation (GD 0.5). Pregnant females at GD 12.5 were anesthetized and received 500 μL deuterium oxide dispersion of 2FT-oCB dots (1 mg/mL) via intrauterine perfusion by 26 G I.V. catheter (Jiangxi Fenglin, China) after abdominal hair removal for the next in vivo fluorescence imaging.

0.5 mg/kgBW LPS (Sigma–Aldrich, USA) was injected into the uteruses of the pregnant mice by intrauterine perfusion at GD 12.5 to induce abortion under anesthesia. The survival fetus and residual tissue were then houexamined by fluorescence hysterography after intrauterine perfusion of 2FT-oCB dots (1 mg/mL, 200 μL, deuterium oxide dispersion) at GD 18.5.

## Reporting summary

Further information on research design is available in the Nature Portfolio Reporting Summary linked to this article.

## Data availability

All data generated or analyzed during this study are included in this published article, Supplementary Information and Source Data file. The X-ray crystallographic data related to 2FT-*o*CB have been deposited at the Cambridge Crystallographic Data Centre (CCDC), under deposition number 2270673. These data can be obtained free of charge from the CCDC via www.ccdc.cam.ac.uk/data_request/cif. There are no restrictions on data availability in the current work. Besides, Source data are provided with this paper.

## Code availability

All codes in this paper are available upon request to the authors.

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

## Acknowledgements

This work was supported by the National Natural Science Foundation of China (61975172 and 82001874), Fundamental Research Funds for the Central Universities (2020-KYY-511108-0007), China Postdoctoral Science Foundation funded project (BX20220260), and Dr. Li Dak Sum & Yip Yio Chin Development Fund for Regenerative Medicine, Zhejiang University. The authors thank Chenyu Yang in the Center of Cryo-Electron Microscopy (CCEM), Zhejiang University for her technical assistance on Transmission Electron Microscopy.

## Author contributions

J.Q., Z.F. and Y.L. conceived and designed the project. Y.L., and Jian.Z. performed the synthesis and theoretical calculations of molecules. J.L. performed the single-crystal analysis. Z.F., S.C., Y.Y., Jun.Z., Y.Z., and X.Y. performed the mouse experiments. Z.F., and S.C. performed the imaging experiments. Y.Y. and Jun.Z. conducted the biocompatibility experiments. Z.F., S.C., and T.W. performed the imaging analysis. Z.F. and Y.L. wrote the original manuscript. Z.F. and J.Q. wrote and reviewed the final version of the text. D.Z., X.Y., and X.F. took part in the discussion and gave important suggestions. J.Q., and B.Z.T. supervised the research.

## Competing interests

The authors declare no competing interests.
