## [Peer Review File · Nature Communications]

Engineered NIR-II fluorophores with ultralong-distance molecular packing for high-contrast deep lesion identificationREVIEWER COMMENTS

Reviewer #1 (Remarks to the Author):

In this work, the authors proposed a design strategy for long-wavelength NIR-II fluorophores by reducing the D-A distance and adjusting the donor. They conducted the X-ray crystallography and gave the single-crystal analysis of the designed 2FT-oCB molecule with the extremely long packing distance of over 8 Å. The as-developed 2FT-oCB dots possess good optical performance. They also studied the biocompatibility of the 2FT-oCB dots in detail, which indicated the dots have good biosafety. In addition, they selected the imaging window delicately using simulation and intravital imaging. After heavy water re-dispersion, they did high-performance in vivo imaging with deep penetration in the highly scattering tissues. This work conveys novel concepts of both NIR-II molecule design and deep in vivo imaging techniques and achieves many advanced results. I believe the authors have done valuable and meaningful jobs and the article is well organized. Thus, I recommend that this paper can be published after the following revisions:

1. In line 130-133, they quantified the molar extinction coefficients of the four molecules at 793 nm which is the wavelength of the excitation light in the imaging applications. However, it is not convincing, since the selected coefficients could not objectively describe the molecular properties. To clearly compare the absorption strength of the four molecules, the authors should give the peak molar extinction coefficients.

2. In line 141-142, the sentence "molecular dyes always suffer fluorescence quenching dispersed from good (organic solvents) to poor solvent (water)" is not correct. The "molecular dyes" is a wide concept in general, which covers the inorganic probes and does not specifically refer to the organic fluorophores. The fluorescence quenching of the small organic molecules in high concentration or aggregated state should be described accurately.

3. Figure 3a & 3b gives the simulation results using the tailing emission of the 2FT-oCB dots in water to select the best imaging window. But the following imaging applications were conducted using the heavy water solution, in which the fluorophores possessed a stronger emission tail. The authors are suggested to perform the imaging simulation again using the emission properties of the dots dispersed in the heavy water.

4. The fluorescence colonography, cystography, and hysterography have been well performed and the imaging results were carefully evaluated. The authors used SSIM to determine whether the non-invasive images are the accurate feedback of the deep tissues, which is an effective and convincing parameter in my opinion. However, they only used SSIM in the bladder image evaluation. Can this method be utilized in the other organ image evaluation?

5. According to Supplementary Fig 32b, the major emission of the 2FT-oCB dots in heavy water beyond 1400 nm seems to be located in the 1400-1500 nm. I am curious about the proportion of the emission within 1400-1500 nm to the emission beyond 1400 nm (1400-1700 nm). Is the emission with 1400-1500 nm strong enough for in vivo imaging? Have the authors tried to perform the in vivo imaging using the 1400-1500 nm window? The light absorption within 1400-1500 nm is relatively higher, which might be beneficial for background suppression. Would it be better?

6. The authors mentioned the re-dispersion in the heavy water finally recovers the bright NIR-II emission and extends it to 1900 nm, which can also be found in Supplementary Fig 32b. However, the current imaging in the work is still below 1700 nm. The authors should clarify the advantages of emission within 1700-1900 nm.

7. Besides the QY beyond 1400 nm, the QY in the whole NIR-II region of the 2FT-oCB dots needs to be given.

8. The components of urine in the bladder are complex. The authors have conducted the fluorescence cystography using the heavy water solution of 2FT-oCB dots, but the stability test of the dots in the urine is missing in the current manuscript.

9. The SBR of the intrauterine foreign body image is extremely high in Fig 5n & 5o, but the power density of excitation light was not provided. It is necessary to make it clear to ensure laser safety.

10. I have found that Fig. 5n and 5o are both the images of intrauterine foreign bodies in the same mouse. Why do the two images exhibit different performances? For example, the left side could not be clearly identified in Fig. 5o but is visible in Fig. 5n. What causes the differences?

11. The authors explored imaging applications including bladder, colon, and uterus imaging, using the heavy water solution of 2FT-oCB dots. Are there any other suitable imaging applications?

12. The fluorophores with peak emission wavelength beyond 1200 nm have been successfully developed in this work by reducing the D-A distance and adjusting the donor. The authors are suggested to give some deep discussion about how to further extend the emission wavelength and improve the emission intensity in the future, aiming for stronger brightness in the window beyond 1400 nm.

13. Besides the designed AIE fluorophores developed in this work, are there any other potential luminophores suitable for biomedical fluorescence imaging within 1400-1700 nm?

14. The scheme of design strategy in Fig. 1a & 1b need to be perfected with much catchier color contrast.

15. Figure 1f gives the AIE curves of the four molecules. The current graphs of the 2MTT-oCB, 2MPT-oCB, and 2FT-oCB need to be enlarged.

16. The authors need to double-check the whole manuscript and figures. For example, the name of the y-axis of Fig. 3a should not be "normalized emission".

Reviewer #2 (Remarks to the Author):

In the manuscript entitled "Engineered NIR-II fluorophores with ultralong-distance molecular packing for over-100-SBR deep lesion identification", the authors focused on the molecular design and optical window selecting for deep bioimaging. Overall, the work is interesting, the study is well designed and the results are clearly presented. This paper is of high quality and may be useful and interesting to chemists, biomedical engineers, etc. However, there still exist some concerns, it can become publishable upon revision, and the existing issues are listed as follows:

1. As shown in figure 2D, the brightness increased from 2MPT-oCB to 2FT-oCB. This step seems also important in the whole design, but is not caused by the emphasized approach of "reducing the D-A distance" in the manuscript. Is it just caused by the donor ability enhancement? It is necessary to explain the reason for this phenomenon from the perspective of molecular design.

2. There exist three measurements using different TH values in figure 4D and figure 4H. The values given in figure 4D and figure 4H is a bit confusing for readers. The authors must carefully recheck them.

3. The description of figure 5N and figure 5O is ambiguous. The imaging conditions are missing. The two images vary to some extent, but lack explanation.

4. The valley value in figure 5O is measured as 0, which may be still confusing for broad readers. The lesions are not labeled by any probes, so they should have no signal logically. Please explain the SBR achievement more fully.

5. Full names should be given when abbreviations appear for the first time in the manuscript, including but not limited to CW, OD, PBS, H&E, etc.

6. Some recent researches involving NIR-II fluorescent probes for bioimaging may be cited, e.g., *Coordination Chemistry Reviews*, 2022, 458, 214438; *Nat. Commun.*, 2021, 12, 6870.

Reviewer #3 (Remarks to the Author):

The present work directs towards developing NIR-II wavelength fluorescent AIEgens (900-1900 nm) for tissue imaging. This is an extension work of the author's previously reported NIR-IIb fluorophore named 2TT-oC26B, which could be excited at 1000 nm with an emission window beyond 1500 nm (*Nature Communications* 2020, 11, 1255). In this work, author have reported small organic molecule based AIEgens beyond 1500 nm by cropping pi-bridge, enhancing donor ability, and by reducing donor-acceptor distance-enabled images of tissue of thickness 4-6 mm with SBR ratio over 100. The synthesized fluorene-based diamine 2FT-oCB exhibited twisted,

ultralong molecular packing distance about 8 Å with tailing emission beyond 1900 nm suppressed imaging background. The present work is well-organized and contributes well to tissue imaging and imaging-guided surgery.

Additional comments:

- 1) Author should comment on the reason behind 2FT-oCB dots dispersion in heavy water (D₂O), causing increased emission intensity, emission shift and regained emission loss beyond 1450 nm when compared with water (H₂O). Is the Absorbance of water beyond 1300 nm is the sole reason for the non-emission of 2FT-oCB in water (Fig. 32a)?
- 2) Pharmacokinetics renal clearance and in vivo stability studies of 2FT-oCB with time course experiment are expected after intravenous injection.
- 3) Reader will expect SBR of whole-body imaging for in-vivo distribution, vascular structures in living mice intravenously injected 2FT-oCB at certain time intervals compared with commercial or FDA-approved NIR dyes such as Indocyanine green (ICG) or any suitable dye.
- 4) The representative 2FT-oCB probe needs to be better characterized. The NMR spectra data is not interpreted and assigned in the Supplementary information (Fig 13 and 14).

Reviewer #1

General Comments: *In this work, the authors proposed a design strategy for long-wavelength NIR-II fluorophores by reducing the D-A distance and adjusting the donor. They conducted the X-ray crystallography and gave the single-crystal analysis of the designed 2FT-oCB molecule with the extremely long packing distance of over 8 Å. The as-developed 2FT-oCB dots possess good optical performance. They also studied the biocompatibility of the 2FT-oCB dots in detail, which indicated the dots have good biosafety. In addition, they selected the imaging window delicately using simulation and intravital imaging. After heavy water re-dispersion, they did high-performance in vivo imaging with deep penetration in the highly scattering tissues. This work conveys novel concepts of both NIR-II molecule design and deep in vivo imaging techniques and achieves many advanced results. I believe the authors have done valuable and meaningful jobs and the article is well organized. Thus, I recommend that this paper can be published after the following revisions:*

Our reply: We appreciate the reviewer for the positive feedback and providing valuable insights that have helped us to considerably improve the quality of our work.

< Comment 1 >

In line 130-133, they quantified the molar extinction coefficients of the four molecules at 793 nm which is the wavelength of the excitation light in the imaging applications. However, it is not convincing, since the selected coefficients could not objectively describe the molecular properties. To clearly compare the absorption strength of the four molecules, the authors should give the peak molar extinction coefficients.

Our reply and modifications:

We appreciate this reviewer for the very helpful suggestion. In order to show the absorption strength of the four molecules clearly, we have added the peak molar extinction coefficients in **Fig. L1 (Supplementary Fig. 18 in the revised Supplementary Information)**.

Fig. L1. The spectra of the four AIEgens in THF showing the molar extinction coefficients at various wavelengths (especially the molar extinction coefficients at 793 nm and the peak molar extinction

coefficients).

Correspondingly, we have made the following modification in our revised manuscript:

“2TT-*o*CB, 2MTT-*o*CB, 2MPT-*o*CB and 2FT-*o*CB show generally increased maximal absorption wavelength at 695, 736, 860 and 829 nm, respectively (see Fig. 1d), with respective peak ϵ of 1.1×10^4 , 1.3×10^4 , 1.8×10^4 , and 2.3×10^4 M⁻¹ cm⁻¹ and ϵ of 0.5×10^4 , 1.0×10^4 , 1.4×10^4 , and 2.2×10^4 M⁻¹ cm⁻¹ at the biological window of 793 nm (see Supplementary Fig. 18).” (Line 3-7, Page 4)

< Comment 2 >

In line 141-142, the sentence “molecular dyes always suffer fluorescence quenching dispersed from good (organic solvents) to poor solvent (water)” is not correct. The “molecular dyes” is a wide concept in general, which covers the inorganic probes and does not specifically refer to the organic fluorophores. The fluorescence quenching of the small organic molecules in high concentration or aggregated state should be described accurately.

Our reply and modifications:

We thank the reviewer for the kind reminder. We have modified the description as follows in our revised manuscript:

“Generally, small organic molecules always suffer fluorescence quenching dispersed from good (organic solvents) to poor solvent (water).” (Line 17-18, Page 4)

< Comment 3 >

*Figure 3a & 3b gives the simulation results using the tailing emission of the 2FT-*o*CB dots in water to select the best imaging window. But the following imaging applications were conducted using the heavy water solution, in which the fluorophores possessed a stronger emission tail. The authors are suggested to perform the imaging simulation again using the emission properties of the dots dispersed in the heavy water.*

Our reply and modifications:

We thank the reviewer for this helpful recommendation. We do find it valuable to use the emission properties of the dots dispersed in the heavy water in imaging simulation. The simulation results shown in **Fig. L2 (Supplementary Fig. 37** in the revised Supplementary Information) have been added in our revised manuscript, which also proving the superiority of the NIR-IIx + NIR-IIb window.

Fig. L2. The imaging simulation of the 2FT-*o*CB dots (heavy water solution) in deep tissues *via* Monte Carlo method.

Correspondingly, we have added the following description in our revised manuscript:

“Considering the emission recovery of the 2FT-*o*CB dots from water to heavy water, we again simulated the imaging through the deep tissues in three windows using the emission properties of the heavy water solution of 2FT-*o*CB dots. The results shown in Supplementary Fig. 37 confirmed the background suppression strength of the NIR-IIx + NIR-IIb window.” (Line 31-34, Page 7)

< **Comment 4** >

The fluorescence colonography, cystography, and hysteroigraphy have been well performed and the imaging results were carefully evaluated. The authors used SSIM to determine whether the non-invasive images are the accurate feedback of the deep tissues, which is an effective and convincing parameter in my opinion. However, they only used SSIM in the bladder image evaluation. Can this method be utilized in the other organ image evaluation?

Our reply and modifications:

We thank the reviewer for this important comment. SSIM assists to evaluate the consistency of the two images with and without tissues disturbance. To catch the original image, we need to open the abdomen of the mouse. However, except for the bladders, the structures and profiles of colons and uteruses would inevitably change after the surgery, which makes it difficult to obtain the original morphology characteristics of the labeled hollow organs. So, we have not utilized the SSIM analysis in all image evaluation processes. In addition, it is believed that the image processing including the binarization and the subsequent area SBR measurements could be one general and efficient method to assess the image quality of the hollow organs. We sincerely hope that the reviewer can agree with us.

In order to give more clarification, we have added the following explanations in our revised manuscript:

“Except for the bladders, the structures and profiles of colons and uteruses would inevitably change after opening the abdomen, which makes it difficult to obtain the original morphology characteristics. So, the SSIM analysis has not been utilized in all image evaluation processes.” (Line 38-41, Page 11)

< Comment 5 >

According to Supplementary Fig 32b, the major emission of the 2FT-oCB dots in heavy water beyond 1400 nm seems to be located in the 1400-1500 nm. I am curious about the proportion of the emission within 1400-1500 nm to the emission beyond 1400 nm (1400-1700 nm). Is the emission with 1400-1500 nm strong enough for *in vivo* imaging? Have the authors tried to perform the *in vivo* imaging using the 1400-1500 nm window? The light absorption within 1400-1500 nm is relatively higher, which might be beneficial for background suppression. Would it be better?

Our reply and modifications:

We thank the reviewer for this constructive suggestion. After calculation, we have found that the emission within 1400-1500 nm (NIR-IIx) accounts for 16.1% of the emission within 1400-1700 nm (NIR-IIx + NIR-IIb) in the water solution, while the proportion rises sharply up to 63.0% after heavy water redispersion. Though the proportion in water solution is not so high, the collection with NIR-IIx extra still enhance the imaging performance, according to Fig. 3 in the manuscript. In fact, the NIR-IIx emission in water-based solution of 2FT-oCB dots could also be utilized for *in vivo* imaging, which is shown in Fig. L3 (Supplementary Fig. 25 in the revised Supplementary Information). Compared with the NIR-IIx + NIR-IIb imaging, the NIR-IIx imaging gives higher contrast in the visualization of superficial blood vessels. However, excellent quality of NIR-IIx imaging may sacrifice partial deep details, since the strong light absorption can simultaneously restrain the useful signals when suppressing the imaging background. In the imaging with rich vertical information, the NIR-IIx + NIR-IIb imaging with large depth of field is a reasonable choice, which can also better utilize the photoluminescence of the 2FT-oCB dots, and bring more signal as well as less background to the NIR-IIb imaging.

Fig. L3. The *in vivo* vessel imaging using water-based solution of the 2FT-oCB dots. The images in (a) 1400-1700 nm window and (b) 1400-1500 nm window. Scale bar, 10 mm. (c) The cross-sectional fluorescence intensity profiles along the yellow line in (a). (d) The cross-sectional fluorescence intensity profiles along the yellow line in (b). The numbers show the SBRs.

In our revised manuscript, we added the imaging results of the mouse vessels in the NIR-IIx (1400-1500 nm) window. Correspondingly, we have added the following descriptions in the revised manuscript:

“The imaging performance of the NIR-IIx window using the water-based solution of the 2FT-oCB dots is also explored, which can be seen in Supplementary Fig. 25. Compared with the NIR-IIx + NIR-IIb imaging, the NIR-IIx imaging gives higher contrast in the visualization of superficial blood vessels. However, excellent quality of NIR-IIx imaging may sacrifice partial deep details, since the strong light absorption can simultaneously restrain the useful signals when suppressing the imaging background. In the imaging with rich vertical information, the NIR-IIx + NIR-IIb imaging with large depth of field is a reasonable choice, which can also better utilize the photoluminescence of the 2FT-oCB dots, and bring more signal as well as less background to the NIR-IIb imaging.” (Line 8-16, Page 6)

“After calculation, the proportion of the NIR-IIx emission to the NIR-IIx + NIR-IIb emission rises sharply, up to 63.0% from 16.1% after heavy water redispersion.” (Line 17-18, Page 7)

< Comment 6 >

The authors mentioned the re-dispersion in the heavy water finally recovers the bright NIR-II emission and extends it to 1900 nm, which can also be found in Supplementary Fig 32b. However, the current imaging in the work is still below 1700 nm. The authors should clarify the advantages of emission within 1700-1900 nm.

Our reply and modifications:

It is a very good question and we thank the reviewer for the comment. The imaging windows of the mainstream in vivo NIR imaging studies are below 1700 nm now, since most commercial InGaAs 2D detectors are highly sensitive within the spectral region of 900-1700 nm. In our previous work (*Light Sci. Appl.*, 2021, 10, 197), the 1700-1880 nm has been defined as the NIR-IIc window and recognized as another potential imaging window. In this window, the light absorption is moderate and the photon scattering is further restrained compared with the windows below 1700 nm. The simulation results of NIR-IIc imaging *via* the Monte Carlo method indicate that the NIR-IIc window and the NIR-IIb window show similar imaging quality in the tissues most containing water. Furthermore, the NIR-IIc imaging exhibits superiority in the adipose tissues due to the elevated light absorption beyond 1700 nm. Thus, we believe the extended window within 1700-1900 nm can widen the spectral band with high imaging performance and also provides new ideas for deep imaging in the fat samples.

Besides, we have noticed that two recent works have realized the in vivo imaging beyond 1700 nm using the SNSPD and MCT detectors. The one study from Professor Hongjie Dai's group (*Nat. Nanotechnol.*, 2022, 17, 653–660) reveals that the photons within NIR-IIb and NIR-IIc windows have the strength of deep penetration, benefit for deep confocal microscopy. In the other study from Professor Yulei Chang's group (*Nat. Commun.* 2023, 14, 1079), the nanoparticles possess strong emission near the window of 1850-1880 nm where the light absorption increases, leading to the efficient background suppression. These two works also convey courage and confidence about the imaging beyond 1700 nm.

Encouraged by the reviewer, we have added the following description in the revised manuscript:

“It needs to be mentioned that the 1700-1880 nm with moderate light absorption and restrained photon scattering has been recognized as the NIR-IIc subwindow²⁵. The long-wavelength emission tailing of 2FT-oCB dots in the NIR-IIc region is beneficial for deep penetration especially through the adipose tissues.” (Line 28-31, Page 11)

References

[25] Feng, Z., *et al.* Perfecting and extending the near-infrared imaging window. *Light: Science & Applications* **10**, 197 (2021).

< Comment 7 >

Besides the QY beyond 1400 nm, the QY in the whole NIR-II region of the 2FT-oCB dots needs to be given.

Our reply and modifications:

We thank the reviewer for the question. In the QY test, we chose IR-26 in the 1,2-dichloroethane (0.5%) as the reference, and the QY beyond 900 nm is calculated as ~0.95% while the QY beyond 1400 nm is further determined as ~0.11%.

The detailed methods were given in the Supplementary Information and we have made the following modification in the revised manuscript:

“With the IR-26 in the 1,2-dichloroethane (0.5%) chosen as the reference (see Fig. 2e), the QY of 2FT-oCB dots beyond 900 nm can be calculated as ~0.95%. Furthermore, the QY beyond 1400 nm can be determined as ~0.11%.” (Line 12-14, Page 5)

< Comment 8 >

The components of urine in the bladder are complex. The authors have conducted the fluorescence cystography using the heavy water solution of 2FT-oCB dots, but the stability test of the dots in the urine is missing in the current manuscript.

Our reply and modifications:

We thank the reviewer for this helpful suggestion. To determine the stability of the 2FT-oCB dots in the urine, we have added the 2FT-oCB dots into the urine collected from mice. As shown in **Fig. L4 (Supplementary Fig. 49)** in the revised Supplementary Information, the fluorescence of 2FT-oCB dots is stable in 60 minutes.

Fig. L4. The stability test of the 2FT-oCB dots in mouse urine. Each sample was measured three times. Error bars indicate SEM (n = 3).

Correspondingly, we have added the following description in the revised manuscript:

“Besides, the good stability of 2FT-*o*CB dots in the urine was confirmed, which could be found in Supplementary Fig. 49.” (Line 16-18, Page 9)

< **Comment 9** >

The SBR of the intrauterine foreign body image is extremely high in Fig 5n & 5o, but the power density of excitation light was not provided. It is necessary to make it clear to ensure laser safety.

Our reply and modifications:

We thank the reviewer for noting the issue. In fact, in uterus imaging, the power density of 793 nm CW laser irradiation is ~ 120 mW/cm², which was list in **Supplementary Table 2**. According to the maximum permissible exposure recommended by the American National Standards Institute (#13997), the power density is below the safety threshold.

According to the reviewer's suggestion, we have added the following description in the revised manuscript:

“It is worth noting that the power density (~ 120 mW/cm²) of the 793 nm CW laser irradiation is below the safety threshold, according to the maximum permissible exposure recommended by the American National Standards Institute (#13997).” (Line 8-10, Page 11)

< **Comment 10** >

I have found that Fig. 5n and 5o are both the images of intrauterine foreign bodies in the same mouse. Why do the two images exhibit different performances? For example, the left side could not be clearly identified in Fig. 5o but is visible in Fig. 5n. What causes the differences?

Our reply and modifications:

We thank the reviewer for noting the confusing descriptions. Figures 5n and 5o show the images of the same mouse with different positions. Due to varying degrees of tissue occlusion, the two sides of the labeled uteruses in the image are distinguishable. Besides, the intrauterine foreign bodies to be identified are not completely parallel to the posture of the mouse, leading to inaccurate identifications of the right uterus in Fig. 5n and the left uterus in Fig. 5o. After slight auxiliary pressing and appropriate position adjusting, the tissue obstruction of the right uterus could be ameliorated and the size of the fetus could be measured.

To clarify the difference between the two images, we have added the description in the revised manuscript:

“After slight auxiliary pressing and appropriate position adjusting, the fetus in the right uterus becomes clearer from Fig. 5n and Fig. 5o due to the amelioration of the tissue obstruction.” (Line 42-44, Page 10)

< **Comment 11** >

*The authors explored imaging applications including bladder, colon, and uterus imaging, using the heavy water solution of 2FT-*o*CB dots. Are there any other suitable imaging applications?*

Our reply and modifications:

Indeed, it is believed that the perfusion of the heavy water solution of 2FT-*o*CB dots can be utilized in many structural tissue or organ imaging including but not limited to the bladder, colon, and uterus. In this work, the heavy water redispersion can efficiently recover the NIR-II signals of bright photoluminescence. On the other hand, the collection of the NIR-IIx window with moderate light absorption could suppress the imaging background. The imaging performance could be thus improved. Furthermore, organ perfusion rather than direct intravenous injection can induce less safety concern, which consequently makes it a general method for hollow organ imaging with translational potential. For example, here we show another case of bile duct imaging using the heavy water solution of 2FT-*o*CB dots, which can be found in **Fig. L5**.

Precise recognition of biliary anatomy is meaningful for the theranostics of bile duct-related complications. After retrograde injection of the heavy water solution of 2FT-*o*CB dots, fluorescence cholangiography was performed in rats. Under the excitation of the 793 nm CW laser, the bile duct images with the abdomen opened were recorded within different imaging windows. As shown in Fig. L5, the imaging windows with moderate light absorption, such as 1200LP, 1300LP, 1400LP, and 1500LP, give restrained outline diffusion and suppressed imaging background. Since the abdomen was opened, the better imaging performance might be mainly owing to the minimized autofluorescence decrease from 1200 nm. Along the yellow line we selected, the results of FWHM analyses reveal that imaging beyond 1200 nm provides better spatial resolutions of below 1 mm among which the NIR-IIx + NIR-IIb image with excellent background suppression gives a minimal FWHM of 0.69 mm in this case.

Fig. L5. NIR-II fluorescence cholangiography within different optical filters. (a-g) The images of (a) 900 LP, (b) 1000 LP, (c) 1100LP, (d) 1200LP, (e) 1300LP, (f) 1400LP, and (g) 1500LP collection. Scale bar, 10 mm. (h-n) The FWHM analyses of the (h) 900 LP, (i) 1000 LP, (j) 1100LP, (k) 1200LP, (l) 1300LP, (m) 1400LP, and (n) 1500LP images. Power density: (a) ~ 10 mW/cm²; (b) ~ 10 mW/cm²; (c) ~ 20 mW/cm²; (d) ~ 20 mW/cm²; (e) ~ 30 mW/cm²; (f) ~ 40 mW/cm²; (g) ~ 40 mW/cm².

< **Comment 12** >

The fluorophores with peak emission wavelength beyond 1200 nm have been successfully developed in this work by reducing the D-A distance and adjusting the donor. The authors are suggested to give some deep discussion about how to further extend the emission wavelength and improve the emission intensity in the future, aiming for stronger brightness in the window beyond 1400 nm.

Our reply and modifications:

We thank the reviewer for this meaningful suggestion. Our work mainly points out that the reduction of the D-A distance for the enhancement of D-A interactions could be efficient for the design of AIE fluorophores with ultralong-wavelength emission. In addition, it has been verified the donor adjusting with increased molecular planarity led to larger absorption and stronger emission from 2MPT-*o*CB to 2FT-*o*CB, though the peak absorption/emission wavelength is hypsochromic shifted. Thus, further incorporating the planar blocks on the twisted backbone could be an efficient approach aiming at bright long-wavelength emission, which has been practiced in our previous work (*ACS Nano*, 2020, **14**, 10, 14228–14239).

On the other hand, some excellent works have revealed that changing the acceptors is an optional strategy for fluorescent emission adjustment, including using the heavy atom effect (*Small* 2019, **15**, 1805549), adding electron-withdrawing substituents (*Adv. Mater.*, 2012, **24**, 2186–2190), coupling with thienothiophene (*Chem. Commun.*, 1995, **22**, 2309–2310), introducing fluorine atom (*Angew. Chem. Int. Ed.*, 2020, **59**, 21049–21057), adding the acceptor numbers (*ChemRxiv*, doi: 10.26434/chemrxiv-2022-89q4q-v3), etc. Further long-wavelength fluorescence optimization could also be focused on acceptor engineering in the future in our opinion, such as replacing the BBTD core and inducing the Se atoms.

As recommended, we have added the following discussion in the revised manuscript:

“In the future, incorporating the planar blocks and adjusting the acceptors (such as replacing the BBTD core and inducing the Se atoms) can be utilized as potential programmes for enhancement of long-wavelength fluorescence emission.” (Line 33-36, Page 11)

< Comment 13 >

Besides the designed AIE fluorophores developed in this work, are there any other potential luminophores suitable for biomedical fluorescence imaging within 1400-1700 nm?

Our reply and modifications:

In addition to the AIE fluorophores we have developed, semiconducting polymers with large conjugation could be potential options for 1400LP fluorescence imaging. Leaving the potential biosafety concern aside, some inorganic nanoluminophores such as near-infrared quantum dots, lanthanide-doped downshifting luminescent nanocrystals, etc., are also excellent candidates for long-wavelength emission. For example, in our previous work, bright quantum dots with peak emission beyond 1400 nm have been utilized for imaging window perfecting and extending (*Light Sci. Appl.*, 2021, **10**, 197). In fact, many inspirational efforts have been made for the biocompatibility improvement of the above-mentioned inorganic nanoprobe (*Angew. Chem. Int. Ed.*, 2020, **59**, 20552; *Nat. Biotechnol.*, 2019, **37**, 1322–1331). Matched to the suitable windows covering the light absorption peak wavelength around 1450 nm, it is believed the biomedical imaging performance would be further upgraded.

To expound our thoughts, we have added the following discussion in the revised manuscript:

“Besides the AIE fluorophores mentioned in this work, some other excellent luminophores, including but not limited to the semiconducting polymers, near-infrared quantum dots, and lanthanide-doped downshifting luminescent nanocrystals, could also be potential candidates for

long-wavelength biomedical fluorescence imaging near the absorption wavelength at around 1450 nm.” (Line 8-12, Page 12)

< Comment 14 >

The scheme of design strategy in Fig. 1a & 1b need to be perfected with much catchier color contrast.

Our reply and modifications:

We thank the reviewer for noting the issue. We have adjusted the colors in Fig. 1a & 1b in the revised manuscript to make them more striking, which is shown in **Fig. L6 (Fig. 1a & Fig. 1b)** in the revised manuscript) below.

Fig. L6. The molecular engineering of NIR-II fluorophores. (a) The donor engineering. (b) The molecular structures and the optical responses.

< Comment 15 >

Figure 1f gives the AIE curves of the four molecules. The current graphs of the 2MTT-oCB, 2MPT-oCB, and 2FT-oCB need to be enlarged.

Our reply and modifications:

We thank the reviewer for this helpful suggestion. The revised image of AIE curves has been given below which is named **Fig. L7 (Fig. 1f)** in the revised manuscript) here. The curves with f_w from 50% to 90% were enlarged in the inserted image to make the AIE characteristics clearly visible.

Fig. L7. The plot of the PL peak intensity of the molecules versus f_w . I and I_0 represent the peak intensities in the mixture with specific f_w , and the pure THF ($f_w = 0$). The inserted image shows the enlarged curves with the f_w from 50% to 90%.

< Comment 16 >

The authors need to double-check the whole manuscript and figures. For example, the name of the y-axis of Fig. 3a should not be “normalized emission”.

Our reply and modifications:

We thank the reviewer for noting the error. The name of the y-axis has been corrected as “Fluorescence intensity (a.u.)”, which could be found in **Fig. L8** (**Fig. 3a** in the revised manuscript).

Fig. L8. The enlarged emission spectrum of 2FT-oCB dots and the absorbance of water in visible-NIR imaging windows.

Reviewer #2

General Comments: *In the manuscript entitled “Engineered NIR-II fluorophores with ultralong-distance molecular packing for over-100-SBR deep lesion identification”, the authors focused on the molecular design and optical window selecting for deep bioimaging. Overall, the work is interesting, the study is well designed and the results are clearly presented. This paper is of high quality and may be useful and interesting to chemists, biomedical engineers, etc. However, there still exist some concerns, it can become publishable upon revision, and the existing issues are listed as follows:*

Our reply: We appreciate the reviewer for the positive feedback, excellent insights, and helpful recommendations.

< Comment 1 >

As shown in figure 2D, the brightness increased from 2MPT-oCB to 2FT-oCB. This step seems also important in the whole design, but is not caused by the emphasized approach of “reducing the D-A distance” in the manuscript. Is it just caused by the donor ability enhancement? It is necessary to explain the reason for this phenomenon from the perspective of molecular design.

Our reply and modifications:

We appreciate receiving this reminder. Besides the adjustment of the donor ability, we believe the increase of the molecular planarity can efficiently improve the molecular absorption strength, thus enhancing the fluorescence brightness. In fact, the approach via incorporating the planar blocks has been successfully practiced (*ACS Nano*, 2020, **14**, 10, 14228–14239). From 2MPT-oCB to 2FT-oCB, the optical responses are blue-shifted (Fig. 1b, 1d & 1e) but the molar extinction coefficients significantly increase (Supplementary Fig. 18). Therefore, the 2FT-oCB dots show remarkably emission enhancement as shown in (Fig. 2d).

To clarify its effectiveness of incorporating the planar blocks, we have modified the following description in our revised manuscript:

“This result indicates that increasing donating ability and cropping the π -bridge can not only redshift the maximal absorption wavelength but also boost the absorption intensity. Especially, though the peak absorption wavelength assumes a slight hypsochromic shift from 2MPT-oCB to 2FT-oCB after adjusting the donor, the absorption strength is efficiently enhanced mainly owing to increasing the planarization.” (Line 7-11, Page 4)

< Comment 2 >

There exist three measurements using different TH values in figure 4D and figure 4H. The values given in figure 4D and figure 4H is a bit confusing for readers. The authors must carefully recheck them.

Our reply and modifications:

We thank the reviewer for noting the confusing values in **Fig. 4d** and **4h**, especially about the statistical results in the bottom half of the two figures. Three TH values were the threshold

values for the signals/background determination in each image binarization. The top half of **Fig. 4d** and **4h** shows the measured areas using different TH values and in the different windows. The area SBRs were calculated with the mean intensities of the segmented image determined as the signals, and the mean intensities of the rest regarded as the background. The bottom half of **Fig. 4d** and **4h** exhibits the statistical results of the segmented images using the three TH values ($n = 3$; Error bars indicate SEM), and the values in columns indicate the mean area SBR. We apologize for the unspecific descriptions in the original manuscript and have made the following modification in the figure captions:

“The statistical analysis is conducted using three segmented images with three TH values ($n = 3$) and the mean area SBRs are shown in the columns.” (Line 7-8, Page 23; Line 2-4, Page 24; Line 5-6, Page 25)

In addition, we have found the mean area SBRs are missing in **Fig. 5c** by accident and the modified **Fig. 5c** in the revised manuscript is shown in **Fig. L9** here.

Fig. L9. The signal areas after binarization and the area SBRs of the uterus images in different spectral regions. The statistical analysis is conducted using three segmented images with three TH values ($n = 3$) and the mean area SBRs are shown in the columns. Error bars indicate SEM.

< **Comment 3** >

The description of figure 5N and figure 5O is ambiguous. The imaging conditions are missing. The two images vary to some extent, but lack explanation.

Our reply and modifications:

We thank the reviewer for noting the confusing presentation, which is also a concern in the Comment 10 of Reviewer #1. Figures 5n and 5o show the images of the same mouse with different positions. Due to varying degrees of tissue occlusion, the two sides of the labeled uteruses in the image are distinguishable. Besides, the intrauterine foreign bodies to be identified are not completely parallel to the posture of the mouse, leading to inaccurate identifications of the right uterus in Fig. 5n and the left uterus in Fig. 5o. After slight auxiliary pressing and appropriate position adjusting, the tissue obstruction of the right uterus could be ameliorated and the size of the fetus could be measured.

To clarify the difference between the two images, we have added the description in the revised manuscript:

“After slight auxiliary pressing and appropriate position adjusting, the fetus in the right uterus becomes clearer from Fig. 5n and Fig. 5o due to the amelioration of the tissue obstruction.”
(Line 42-44, Page 10)

< **Comment 4** >

The valley value in figure 5O is measured as 0, which may be still confusing for broad readers. The lesions are not labeled by any probes, so they should have no signal logically. Please explain the SBR achievement more fully.

Our reply and modifications:

It is a very good question which may be beneficial for us to better demonstrate the advantages of our work. In the window of 1400-1700 nm, the lesions with much water strongly absorb photons and efficiently restrain the background, thus leading to the zero value of intensity. Meanwhile, the 2FT-*o*CB fluorophores exhibit bright NIR-IIx + NIR-IIb emission, which efficiently label the uterus after intrauterine injection. Therefore, the uterine cavity is positively stained and the intrauterine residue is negatively stained. In the analysis of Fig. 5o, the valley value represents the lowest intensity of the residual fetus, while the peak value shows the bright NIR-IIx + NIR-IIb signal labeling of the uterine cavity. When the bright fluorescence intensity reaches 255 (the largest value in the 8-bit image), the measured intensity of lesions is still as low as zero, thus leading to a high contrast approaching positive infinity.

To make it clearer, we have made the following modifications in the revised manuscript:

“Since the residual pregnancy tissues contain much water, the tissue background is efficiently suppressed in the NIR-IIx + NIR-IIb window, leading to strong negative staining. According to the analysis in Fig. 5n, the calculated SBR comes to over 100.” (Line 37-39, Page 10)

“As shown in Fig. 5o, the valley value represents the lowest intensity of the negatively stained fetus, while the peak value shows the bright NIR-IIx + NIR-IIb signal labeling of the uterine cavity.” (Line 44, Page 10; Line 1-2, Page 11)

< **Comment 5** >

Full names should be given when abbreviations appear for the first time in the manuscript, including but not limited to CW, OD, PBS, H&E, etc.

Our reply and modifications:

We thank the reviewer for noting the abbreviations issues in the manuscript.

We have rechecked our manuscript and made the following corrections in the revised manuscript:

“After uterine perfusion of the 2FT-*o*CB dots (1 mg/mL, 200 μ L, deuterium oxide dispersion), the anesthetized mice laid on the imaging stage with the lower abdomen exposed to the irradiation of 793 nm continuous wave (CW) laser.” (Line 27-29, Page 9)

“To directly compare the in vivo performance in different imaging windows and determine the optimal long-pass detection region of NIR-II fluorescence off-peak imaging, the whole-body vessel imaging in mice was then conducted after intravenous injection of 1 \times phosphate-buffered

saline (PBS) solution of 2FT-*o*CB dots (1 mg/mL, 200 μ L, normal water-based buffer solution).” (Line 27-31, Page 5)

“To examine the safety of 2FT-*o*CB dots *in vivo*, two groups of Institute of Cancer Research (ICR) mice were administered with an intravenous dose of AIE dots (as the experimental group) and 1 \times PBS (as the control group), respectively.” (Line 33-36, Page 6)

“The hematoxylin and eosin (H&E) staining results of the bladders in Supplementary Fig. 48 show that there exists no obvious difference between the groups perfused with the heavy water dispersion of 2FT-*o*CB dots and the 1 \times PBS, respectively.” (Line 14-16, Page 9)

Besides, some terms have been abbreviated when they reappear:

“2FT-*o*CB dots can be listed as one of the few fluorophores that simultaneously achieve long-wavelength optical responses, strong light absorption, and high QY beyond 1400/1500 nm.” (Line 5-7, Page 12)

“A solution containing 3 mg of 2TT-*o*CB/2MTT-*o*CB/2MPT-*o*CB/2FT-*o*CB, 15 mg of Pluronic F-127, and 1.5 mL of THF was added into 10 mL of deionized water.” (Line 21-22, Page 12)

“The BALB/c nude mice (~20 g) and ICR mice (~20 g) involved in this work were provided from the SLAC Laboratory Animal Co. Ltd. (Shanghai, China).” (Line 31-32, Page 12)

“(d) The normalized absorption and (e) the normalized PL spectra of the four molecules dissolved in THF.” (Line 5-6, Page 19)

Furthermore, some other corrections in the revised Supplementary Information are as follows:

“The intensities were plotted as a function of optical density (OD), respectively, and the slopes were calculated by linear fittings.” (Line 22-23, Page 3)

< Comment 6 >

Some recent researches involving NIR-II fluorescent probes for bioimaging may be cited, e.g., Coordination Chemistry Reviews, 2022, 458, 214438; Nat. Commun., 2021, 12, 6870.

Our reply and modifications:

We thank the suggestions. In our revised manuscript, we have added the following citations:

“Recently, the second near-infrared (NIR-II, 900-1880 nm) window gives a high-fidelity preservation for deep signals’ detection *in vivo*³⁻¹⁰.” (Line 1-2, Page 2)

“Aggregation-induced emission (AIE) effect could be an efficient strategy in parallel to manufacture bright long-wavelength-emitting organic fluorophores⁴⁸⁻⁵³.” (Line 20-22, Page 4)

References

[10] Chen, J., *et al.* A H₂O₂-activatable nanoprobe for diagnosing interstitial cystitis and liver ischemia-reperfusion injury via multispectral optoacoustic tomography and NIR-II fluorescent imaging. *Nature Communications* **12**, 6870 (2021).

[53] Ouyang, J., Sun, L., Zeng, F. & Wu, S. Biomarker-activatable probes based on smart AIEgens for fluorescence and optoacoustic imaging. *Coordination Chemistry Reviews* **458**, 214438 (2022).

Reviewer #3

General Comments: *The present work directs towards developing NIR-II wavelength fluorescent AIEgens (900-1900 nm) for tissue imaging. This is an extension work of the author's previously reported NIR-IIb fluorophore named 2TT-oC26B, which could be excited at 1000 nm with an emission window beyond 1500 nm (Nature Communications 2020, 11, 1255). In this work, author have reported small organic molecule based AIEgens beyond 1500 nm by cropping pi-bridge, enhancing donor ability, and by reducing donor-acceptor distance-enabled images of tissue of thickness 4-6 mm with SBR ratio over 100. The synthesized fluorene-based diamine 2FT-oCB exhibited twisted, ultralong molecular packing distance about 8 Å with tailing emission beyond 1900 nm suppressed imaging background. The present work is well-organized and contributes well to tissue imaging and imaging-guided surgery.*

Our reply: We appreciate the reviewer for the positive feedback and valuable recommendations that has helped us to improve its quality.

< Comment 1 >

Author should comment on the reason behind 2FT-oCB dots dispersion in heavy water (D2O), causing increased emission intensity, emission shift and regained emission loss beyond 1450 nm when compared with water (H2O). Is the Absorbance of water beyond 1300 nm is the sole reason for the non-emission of 2FT-oCB in water (Fig. 32a)?

Our reply and modifications:

We thank the reviewer for the constructive comment. To reveal the leading role of solvent transference, we calculated the attenuated emission spectrum using the emission spectrum in the heavy water solution and light absorption properties of water, and compared the calculated emission spectrum with the emission spectrum measured in water solution. The comparison results are shown in **Fig. L10 (Supplementary Fig. 36** in the revised Supplementary Information). After tuning the parameters, a fitted attenuation process could be established with 95% confidence bounds and thus the credible simulation data with photon restraining in water could be given (named as calculated data in **Fig. L10**). It could be seen from **Fig. L10** that the severe emission loss beyond 1400 nm is consistent and the blue-shifted peak emission wavelength difference is just 3 nm (1118 nm of the calculated data and 1115 nm of the measured intensity in water, which may be caused by the simulation errors).

Fig. L10. The comparison between the calculated spectrum and the measured spectrum of water solution. The calculated data was simulated *via* a fitted attenuation process with 95% confidence bounds using the emission property of 2FT-*o*CB dots in the heavy water and the absorption spectrum of water. Comparing the calculated data with the measured spectrum in water could directly reveal the leading role of solvent transference in the differences in emission spectra of 2FT-*o*CB dots in the two solvents.

In addition, we recorded the NIR-II image of ICG and 1450QDs (PbS/CdS quantum dots emitting at \sim 1450 nm) in heavy water and water, as shown in **Fig. L11**. Similar emission losses beyond 1000 nm and 1400 nm have been detected. Therefore, it is believed NIR-II light absorption of water plays the main role in the emission loss when transferring the solvent from heavy water to water.

Fig. L11. The NIR-II emission comparison between the heavy water solution and water solution of ICG and 1450QDs. (a) The NIR-II images of ICG in the two solvents beyond 1000 nm and 1400 nm. (b) The quantified intensity of ICG ($n = 3$). (c) The NIR-II images of 1450QDs in the two solvents beyond 1000 nm and 1400 nm. (d) The quantified intensity of 1450QDs ($n = 3$).

Correspondingly, we have updated the following description in the revised manuscript:

“We further calculated the attenuated emission results using the emission spectrum of 2FT-*o*CB dots in the heavy water and light absorption properties of water. After tuning the parameters, a fitted attenuation process was established with 95% confidence bounds, and a credible simulation data with photon restraining in water was obtained and shown in Supplementary Fig. 36 with the name of calculated data. Compared with the measured spectrum of 2FT-*o*CB dots in water, it can be seen that the severe emission loss beyond 1400 nm of the calculated data and the measured data is consistent and the blue-shifted peak emission wavelength difference is just 3 nm (1118 nm of the calculated data and 1115 nm of the measured intensity in water, which may be caused by the simulation errors), indicating the main role played by water in the emission differences of 2FT-*o*CB dots in the two solvents.” (Line 21-30, Page 7)

< **Comment 2** >

*Pharmacokinetics renal clearance and in vivo stability studies of 2FT-*o*CB with time course experiment are expected after intravenous injection.*

Our reply and modifications:

As suggested, we evaluated the long-term imaging ability after injection of the 2FT-*o*CB dots, as displayed in **Fig. L12 (Supplementary Fig. 26** in the revised Supplementary Information). It can be seen that 2FT-*o*CB dots possess good stability in the blood circulation system and gradually accumulate in the liver and gut. We also measured the S/B values (the quotient of signal and background) of the hind limbs at each time point and plotted the curve in **Fig. L13**, where the signals were determined in the left hind limb and the backgrounds were selected outside the mouse.

Fig. L12. The whole-body imaging of mice after intravenous injection of 2FT-*o*CB dots within 24 hours. Scale bar, 10 mm.

Fig. L13. The S/B values of the left hind limb at each time point post injection.

The feces and urine detection after intravenous injection of the 2FT-*o*CB dots shown in **Fig. L14 (Supplementary Fig. 28** in the revised Supplementary Information) demonstrates that hepatobiliary excretion is the main route rather than renal excretion.

Fig. L14. The NIR-II fluorescence detection of mice feces and urine in one week. The feces and urine from the mice with no treatment were considered as the control group; the feces and urine from the mice after intravenous injection of 2FT-*o*CB dots were regarded as the experimental group. The upper images were taken by the phone and the below images were taken by the InGaAs camera under the excitation of the 793 nm CW laser.

Correspondingly, we have added the following description in our revised manuscript:

“The whole-body imaging of mice was performed within 24 hours after intravenous injection. Besides the liver, the gut is also gradually lighted up, which could be seen in Supplementary Fig. 26.” (Line 19-21, Page 6)

“Meanwhile, the urine excreted by the mice treated with 2FT-*o*CB dots shows no difference in the fluorescence detection compared with the control groups. These results definitely demonstrate that hepatobiliary excretion is the main route rather than renal excretion.” (Line 27-30, Page 6)

< Comment 3 >

Reader will expect SBR of whole-body imaging for *in-vivo* distribution, vascular structures in living mice intravenously injected 2FT-*o*CB at certain time intervals compared with commercial or FDA-approved NIR dyes such as Indocyanine green (ICG) or any suitable dye.

Our reply and modifications:

We thank the reviewer for the useful suggestion which may further demonstrate the superiority of the newly developed fluorophores. The vascular imaging and *in-vivo* distribution using 2FT-*o*CB dots and ICG dye were given in **Fig. L15a&b (Supplementary Fig. 27a&b** in the revised Supplementary Information). Obviously, 2FT-*o*CB dots possess a longer blood circulation time and give better qualities of vessel imaging within at least 1 h. The imaging beyond 1400 nm using 2FT-*o*CB dots and ICG dye as probes can be seen in **Fig. L15c (Supplementary Fig. 27c** in the revised Supplementary Information). In the excellent NIR-IIx + NIR-IIb imaging window, the 2FT-*o*CB dots also exhibit better contrast.

Fig. L15. The NIR-II fluorescence imaging comparison after intravenous injection of the 2FT-*o*CB dots and ICG. (a) The images of mouse vessels using 2FT-*o*CB dots with 1300LP collection at

various time points post-injection. (b) The images of mouse vessels using ICG with 1300LP collection at various time points post-injection. (c) The imaging comparison at 1 min post-injection in the NIR-IIx + NIR-IIb window. The numbers show the SBRs. Obviously, imaging using 2FT-*o*CB dots possesses better performance. Scale bars, 10 mm.

We have also added the corresponding description in our revised manuscript:

“Compared with the indocyanine green (ICG)-assisted vessel imaging, a longer blood circulation time could be achieved using 2FT-*o*CB dots (see Supplementary Fig. 27a&b). In the excellent 1400-1700 nm window, 2FT-*o*CB dots show better imaging quality than ICG, which could be found in Supplementary Fig. 27c.” (Line 21-24, Page 6)

< Comment 4 >

*The representative 2FT-*o*CB probe needs to be better characterized. The NMR spectra data is not interpreted and assigned in the Supplementary information (Fig 13 and 14).*

Our reply and modifications:

We thank the reviewer for the suggestion. The revised NMR spectra of 2FT-*o*CB are shown in **Fig. L16 (Supplementary Fig. 13** in the revised Supplementary Information) and **Fig. L17 (Supplementary Fig. 14** in the revised Supplementary Information) with necessary assignments.

Fig. L16. ¹H NMR spectrum of 2FT-*o*CB.

Fig. L17. ¹³C NMR spectrum of 2FT-oCB.

REVIEWERS' COMMENTS

Reviewer #1 (Remarks to the Author):

The reviewer carefully read the manuscript, the previous reviewers' comments and authors' point-to-point reply. The authors provided lots of new data to address the reviewers' concerns and carefully revised manuscript accordingly. The research reported in the study represents the major advancement in the field of NIR-II imaging probe development and highly novel. Thus, the reviewer recommends the acceptance of the manuscript for publication in the current form.

Reviewer #2 (Remarks to the Author):

The manuscript has been improved and is ready for publication.

Reviewer #3 (Remarks to the Author):

The revision has improved the quality of the paper and we recommend that it be published as is.

Reviewer #1

The reviewer carefully read the manuscript, the previous reviewers' comments and authors' point-to-point reply. The authors provided lots of new data to address the reviewers' concerns and carefully revised manuscript accordingly. The research reported in the study represents the major advancement in the field of NIR-II imaging probe development and highly novel. Thus, the reviewer recommends the acceptance of the manuscript for publication in the current form.

Our reply: We appreciate the encouragement from Reviewer #1.

Reviewer #2

The manuscript has been improved and is ready for publication.

Our reply: We thank the kindness from Reviewer #2.

Reviewer #3

The revision has improved the quality of the paper and we recommend that it be published as is.

Our reply: We thank the recognition from Reviewer #3.